# The Link Between Oxysterols and Gut Microbiota in the Co-Dysfunction of Cognition and Muscle

**DOI:** 10.3390/nu17071277

**Published:** 2025-04-06

**Authors:** Mengwei Ju, Wenjing Feng, Zhiting Guo, Kexin Yang, Tao Wang, Huiyan Yu, Chengyan Qi, Miao Liu, Jiaxuan Tao, Rong Xiao

**Affiliations:** School of Public Health, Capital Medical University, Beijing 100069, China; meave_ju@163.com (M.J.); 15810888862@163.com (W.F.); guozhiting51@163.com (Z.G.); yangkx5989@163.com (K.Y.); wangtao_930106@163.com (T.W.); bjyuhuiyan@126.com (H.Y.); shunshun006572@163.com (C.Q.); liumiao0330@163.com (M.L.); taojxxx@163.com (J.T.)

**Keywords:** mild cognitive impairment, muscle function, oxysterols, gut microbiota, comorbidity

## Abstract

**Background/Objectives:** Alterations of oxysterols and gut microbiota have been recognized as indicators affecting mild cognitive impairment (MCI) and sarcopenia, respectively, whereas their association with co-dysfunction has not been investigated. **Methods:** In this study, a total of 1035 individuals were divided into Control (*n* = 264), MCI (*n* = 435), and MCI with possible sarcopenia (MPS, *n* = 336) groups. Cognition and muscle indexes, serum oxysterols, and gut microbiota were measured. Spearman’s rank coefficients were calculated to determine their correlations. **Results:** Performances of global and multidimensional cognitive tests was successively worse in the Control, MCI, and MPS groups. Longer duration of five-time chair stand test, lower 6-meter walk speed, and handgrip strength were observed in the MPS group, along with increased 27-hydroxycholesterol (27-OHC) and 5α,6α-epoxycholesterol and decreased 5α-Cholest-8(14)-ene-3β,15α-diol (15-HC). Higher concentrations of amyloid precursor protein (APP), neurofilament, and C-terminal agrin fragment (CAF) were discovered in the MCI and MPS groups. The α-diversity of gut microbiota in the MCI and MPS group was remarkably decreased, followed by a shifted abundance of microbial taxa, such as Alistipes and Rikenellaceae. Multiple significant correlations were found between cognition and muscle indexes and with oxysterols. **Conclusions:** Our study indicates that oxysterols and gut microbiota are prominently involved in the co-dysfunction of cognition and muscle.

## 1. Introduction

Mild cognitive impairment (MCI) has been recognized as the preclinical stage of Alzheimer’s disease (AD) [1]. With the continuous progress of aging, there are growing shreds of evidence that MCI is forcefully bidirectionally correlated with sarcopenia [2,3]. MCI individuals have been reported to present a series of sarcopenia statuses including low muscle mass, declined handgrip strength, decelerated pace, and so on [4,5]. Considering that no effective treatment is available at present, the early identification and prevention of risk factors are crucial.

Given the common risk factors of comorbidity for MCI and sarcopenia, studies have suggested that nutrition is a modifiable indicator [6]. A meta-analysis revealed that higher adherence to the Mediterranean diet was relevant to better physical performance and cognitive function [7]. Specifically, as a recognized accelerator of metabolic diseases, cholesterol has been demonstrated to be involved in the progression of MCI and sarcopenia, respectively, while research on the latter is relatively rare. Our previous research discovered that there exists quadratic and longitudinal relations between dietary cholesterol intake with cognitive function [8]. In addition, our team confirmed that cholesterol intake and lipid profiles were prominently diverse in MCI, sarcopenia, and MCI+ sarcopenia groups [9]. Consequently, dietary cholesterol-related pathways are one of the vital research targets of comorbidity for MCI and sarcopenia. Since cholesterol is incapable of crossing the blood–brain barrier (BBB), its oxygenated derivatives, oxysterols, may play an interfering role in the central nervous system [10]. Among them, 27-hydroxycholesterol (27-OHC), which is the most abundant oxysterol in the periphery, was proven to be an independent risk factor for MCI [11]. Nevertheless, its muscle toxicity has not been explored yet. Furthermore, higher plasma levels of 7β-hydroxycholesterol (7β-OHC) and 7-ketocholesterol (7-KC) were detected in sarcopenic individuals, which were classified as oxidative stress biomarkers of sarcopenia [12]. In conclusion, it is worth exploring whether there is a remarkable correlation between various oxysterols and the co-dysfunction of cognition and muscle.

As age increases, the abundance and diversity of gut microbiota in the elderly changed notably, which were regulated by various environmental factors, including dietary cholesterol [13]. On the one hand, the change in gut microbiota may regulate cognitive function by regulating amyloid-β (Aβ) deposition, Tau phosphorylation, neuroinflammation, metabolic dysfunction, and oxidative stress, which is defined as the gut–brain axis [14,15]. On the other hand, gut microbiota dysbiosis impairs muscle mass and function by modulating muscle protein synthesis, muscle mitochondria function, systemic chronic inflammation, and depressant muscle regeneration, which is the gut–muscle axis [16]. To summarize, gut microbiota may be the intermediate link between cognitive function and muscle performance, whereas there are few studies on the abnormal alteration of gut microbiota in comorbidities.

## 2. Materials and Methods

### 2.1. Participants

The participants, aged 50~75 years, were recruited from a multicenter prospective cohort named Effects and Mechanism Investigation of Cholesterol and Oxysterol on Alzheimer’s Disease (EMCOA). Of these, participants were administered face-to-face questionnaire interviews with the collection of demographic data, chronic disease history, dietary surveys, and neuropsychological testing, as well as muscle quality and functionality. In addition, the fasting blood samples and fresh fecal samples were collected during the interviews and then stored at −80 °C following standardized protocols. The exclusion criteria were as follows: (i) severe diseases such as cancer, the failure of heart, respiratory, renal, or liver; (ii) specific diseases caused neurologic or psychiatric disorders, like traumatic brain injury, epilepsy, and Parkinson’s disease; (iii) taking medications or dietary supplements affecting cognition; and (iv) inability to complete muscle tests due to paralysis or weakness of limbs. Ultimately, a total of 1035 cases were randomly included in this study, comprising 264 cases in the Control group, 435 cases in the MCI group, and 336 cases in the MCI with possible sarcopenia (MPS) group (Figure 1). The study protocol and written informed consent were obtained from all subjects with the approvement of the Ethics Committee of Capital Medical University (No.2013SY35).

To ensure the quality of fecal samples and the reliability of the data, a rigorous protocol was implemented: Study participants were required to provide fasting stool specimens simultaneously with venous blood collection during the designated morning session under standardized fasting conditions (≥10 h). Trained staff provided one-on-one guidance on sample collection using medical-standard containers, requiring a minimum broad bean-sized volume (~2 g) and storage at −80 °C within 2 h of collection. A single freezing protocol was implemented to avoid degradation caused by freeze–thaw cycles.

### 2.2. Diagnostic Criteria of MCI and MPS

A Montreal Cognitive Assessment (MoCA) was carried out to preliminarily screen MCI according to standard protocols. Briefly, subjects reported memory decline during follow-up, and at the same time were eligible for the cutoff scores of MoCA classified by education years: ≤14 for illiterate individuals; ≤19 for those with 1~6 years of education; and ≤24 for those with 7 or more years of education. Afterwards, the final diagnosis was established by neurologists.

Individuals were further measured for muscle function to evaluate whether they complied with the diagnostic criteria of possible sarcopenia according to the consensus of the Asian Working Group for Sarcopenia in 2019 (AWGS2019) [17]. In detail, participants who met one of the following conditions were further diagnosed by orthopedic specialists for possible sarcopenia: (i) handgrip strength: male < 28 kg, female < 18 kg; (ii) five-time chair stand test ≥ 12 s. Eventually, MCI subjects with possible sarcopenia were classified as the MPS group; otherwise, individuals were included in the MCI group.

### 2.3. Dietary Survey and Assessment

A Food Frequency Questionnaire (FFQ) was utilized to appraise dietary information. The semiquantitative FFQ was comprising 33 food items, of which the frequency (day, week, month, year) and amount (in grams or milliliter) were provided separately within 1 year before the survey (details are shown in Appendix A). The FFQ used in this study was provided by the National Institute of Nutrition and Health, Chinese Center for Disease Control and Prevention, and underwent training and validation through preliminary research conducted by our team. The conversion of food into nutrients was calculated (nutrient composition in food × frequency × amounts) with reference to the “Chinese Food Composition Table”.

### 2.4. Cognitive Function Assessment

MoCA and Mini-Mental State Examination (MMSE) were conducted to evaluate global cognitive function. Multidimensional cognitive function was assessed at the same time. Specifically, the Auditory Verbal Learning Tests (AVLTs) were used to estimate verbal memory ability, including short recall (AVLT-SR) and long recall (AVLT-LR). The Symbol Digit Modalities Test (SDMT) was utilized to assess processing speed, while Digit Span Forwards/Backwards (DSF/DSB) and Trail Making Test A/B (TMT-A/TMT-B) were used to assess working memory, executive function, and episodic memory ability. Moreover, the Stroop Color Word Test (SCWT), including SCWT–time-interfered effects (SCWT-TIE) and SCWT–reaction-interfered effects (SCWT-RIE), was the evaluation tool of attention.

### 2.5. Body Composition Test

Under the guidance of investigators, participants stood barefoot at the designated positions of the body composition analyzer (Inbody720, InBody, Seoul, Republic of Korea), gripping the handles of the analyzer with both hands and facing forward. Participants were required to undergo the test with overnight fasting and remove all metal objects they were carrying. Individuals with implanted metal medical devices, such as cardiac pacemakers, were excluded from participating in this test.

### 2.6. Calf Circumference Measurement

Calf circumferences were measured by uniformly trained investigators. The participants were supposed to stand naturally, while investigators wrapped around the thickest part of the calf with a flexible tape measure, moving up and down along the calf to ascertain the maximum circumference. It was ensured that the tape measure adhered to the skin without compressing the calf to avoid measurement errors caused by indentation in the leg. The measurement was repeated three times and the average value was recorded.

### 2.7. Handgrip Strength Test

Participants were instructed to grip the dynamometer (KYTO, Dongguan, China) with their dominant hand and squeeze it with maximum force for 3–5 s at the beginning of the test. Throughout the testing process, participants were required to keep their arms hanging down naturally, without assistance from other parts of the body. The test was repeated three times, with an approximately 1 min interval between each measurement.

### 2.8. The Five-Time Chair Stand Test

The five-time chair stand test was performed to assess physical performance. In simple terms, participants sat on a fixed chair with their feet flat on the floor. Timing started when they stood up, fully extending their knees until they were completely upright before sitting back down, which counted as one repetition. This procedure was repeated five consecutive times, and the total duration was recorded.

### 2.9. Serum Lipids, Oxysterols, and Biomarkers Detection

Serum total cholesterol (TC), triglycerides (TG), high-density lipoprotein cholesterol (HDL-C), and low-density lipoprotein cholesterol (LDL-C) were detected using the Roche Hitachi 8000 C automatic biochemical analysis system (Roche, Basel, Switzerland).

Serum oxysterols were measured using high-performance liquid chromatography–mass spectrometry (HPLC–MS). Briefly, 10 mg of isomeric mixed internal standards powder, composed of d7-24-hydroxycholesterol, d7-7β-hydroxycholesterol, d6-25-hydroxycholesterol, d7-7-ketocholesterol, and d7-7α-hydroxy-4-cholesten-3-one, was diluted with methanol to 0.2 μg/mL as a final concentration. Next, serum samples were mixed with 1% 2,6-di-tert-butylp-cresol (BHT) ethanol solution and internal standard, in which the former played an antioxidant role. After that, samples were incubated at 25 °C, 1200 rpm for 1 h, centrifuged, and extracted for supernatants, which were dried afterward and quantified with the added internal standard. Ultra-performance liquid chromatography-Q-trap-tandem mass spectrometry (QTRAP 6500 PLUS, AB SCIEX, Framingham, MA, USA) was carried out for liquid chromatography-mass spectrometry detection. The mobile phases A and B consisted of (chloroform:methanol:ammonium hydroxide = 89.5:10:0.5) and (chloroform:methanol:ammonium hydroxide:water = 55:39:0.5:5), respectively. The chromatographic elution gradient was as follows: 5% B, 3 min → 5~40% B, 9 min → 40% B, 4 min → 40~70% B, 5 min → 70% B, 15 min → 70~5% B, 3 min → equilibrated for 6 min before the next injection. A Phenomenex Luna Silica 3 µm column (internal diameter 150 × 2.0 mm; column temperature, 25 °C; flow rate, 270 μL/min) was used for the separation of oxysterols with the 5 μL of sample volume. The mass spectrometry parameters were as follows: curtain gas, GAS1, and GAS2 at 20, 35, and 35 psi, respectively; ion source temperature at 400 °C; spray voltage at 5.5 kV. Eventually, the data were collected in electrospray ionization (ESI) mode, while the quantification of oxysterols was performed with the multiple reaction monitoring (MRM) mode.

Concentrations of Aβ1-42 (E-EL-H0543), amyloid precursor protein (APP, E-EL-H1216), neurofilament (Nfl, E-EL-H0741c), brain-derived neurotrophic factor (BDNF, E-EL-H0010c), C-terminal agrin fragment (CAF, ELK9769), and irisin (E-EL-H5735c) were evaluated using an ELISA kit (Arco Biosystems, Newark, DE, USA). Detailed experimental steps are shown in the Appendix A.

### 2.10. Fecal Gut Microbiota

16S ribosomal DNA (rDNA) gene sequencing was performed to characterize the diversity and community composition of gut microbiota, which was fully described previously [18]. In brief, the DNA of fecal samples was extracted using the CTAB method, followed by amplification of V4 region (F515/R806) of 16S rDNA. The Quantitative Insights Into Microbial Ecology2 pipeline was used to analyze the sequences, of which those that were high-quality were clustered into amplicon sequence variants (ASVs) when the similarity threshold matched 100%. On this basis, the indexes of Shannon, Chao1, and Simpson were calculated to estimate α-diversity, while Kruskal–Wallis tests were performed to discover the significance of indexes among groups. At the same time, β-diversity was assessed using a principal component analysis (PCA). To distinguish the bacterial taxa of driving differences between groups, linear discriminatory analysis effect size (LEfSe) was conducted by default criteria (*p* < 0.05 using a Kruskal–Wallis test along with a linear discriminant analysis score > 3).

### 2.11. Statistical Analysis

All statistical analyses were conducted using IBM SPSS version 26.0 (IBM Corporation, Armonk, NY, USA) software, R software (version 3.6.1, R Foundation for Statistical Computing), and GraphPad Prism 7.0.0 software. Continuous variables were presented as mean ± standard deviation or median (interquartile range), while categorical variables as frequencies (percentages). All data were analyzed using one-way analysis of variance (ANOVA), Kruskal–Wallis rank sum test, chi-square test, or Fisher’s exact test to determine group differences. Spearman’s rank coefficients were analyzed to determine the correlation between oxysterols with cognition and muscle indexes. A two-sided *p* value < 0.05 was considered statistically significant.

## 3. Results

### 3.1. Participants Characteristics

The characteristics of demographic, serum lipids, and dietary nutrient intake are presented in Table 1. Of the 1035 participants, the amounts in the Control, MCI, and MPS groups are 264, 435, and 336, respectively. In detail, older age was observed in the MPS group compared with the Control and MCI groups (*p* < 0.001). In addition, there were more subjects with <9 years of education but fewer subjects with >12 years in the MPS group than the Control group (*p* < 0.001). With significant differences, here we found that the MPS group showed shorter height compared with the Control group (*p* = 0.035). No remarkable difference was discovered in regard to gender, weight, BMI, smokers, and drinkers (*p* > 0.05).

Notably, MPS individuals were more likely to have lower concentrations of serum HDL-C than the Control group (*p* = 0.001), while no prominent diversity was found in TC, TG, and LDL-C. Similarly, participants among groups were supposed to intake relatively consistent dietary nutrients including energy, protein, fat, and cholesterol (*p* > 0.05).

### 3.2. Cognitive Test Scores

A series of global and multidimensional test scores were calculated to evaluate the cognitive functions of subjects. As shown in Table 2, most tests were different among the three groups. Specifically, the scores of MMSE, MoCA, AVLT-SR, AVLT-LR, SDMT, and DSTB were decreased sequentially in the Control, MCI, and MPS groups (all *p* < 0.001). In addition, the times of TMT-A and TMT-B were elevated in turn (*p* < 0.001). All of the above results point to the conclusion that global and multidimensional cognitive functions decrease in a sequential manner with MCI and MPS.

A gender-stratified analysis was conducted to investigate differential cognitive performance patterns. In female participants, the Control, MCI, and MPS groups exhibited statistically significant stepwise declines across multiple cognitive assessments (MMSE, MoCA, AVLT-SR, AVLT-LR). This graded pattern was not observed in males, where only reduced performance in both MCI and MPS groups relative to Controls reached significance. The differences in DSF and TMT-B performance observed in both males and females mirrored the patterns identified in the general population analysis. Notably, male participants demonstrated additional distinct characteristics: the MPS group showed significantly lower DSB and SDMT scores compared to both MCI and Control groups, while displaying elevated TMT-A scores. In contrast, female participants exhibited progressive cognitive decline patterns, with both MCI and MPS groups demonstrating reduced DSB performance compared to Controls, and SDMT scores showing a significant stepwise decrease across the three groups.

Age-stratified analyses demonstrated distinct cognitive patterns while maintaining consistency with overall trends. Both age subgroups (<65 and ≥65 years) exhibited significant progressive declines in SDMT scores accompanied by progressive elevation in TMT-B performance, mirroring population-level observations. For MoCA and AVLT-SR assessments, while the MCI and MPS groups showed reduced performance compared to controls across age strata, no significant differences emerged between the MCI and MPS groups themselves. Notably, age-specific divergences were observed: In the <65 y subgroup, DSB displayed significant stepwise deterioration, with concurrent reductions in MMSE, AVLT-LR, and DSF scores in both MCI and MPS groups relative to Controls, while TMT-A scores were specifically elevated in the MPS group. Conversely, the ≥65 y subgroup revealed a progressive improvement in MMSE and AVLT-LR scores across diagnostic groups, contrasted by significant stepwise worsening in TMT-A performance. Both MCI and MPS groups maintained reduced DSB compared to Controls in older adults, whereas DSF deficits became exclusive to the MPS group in this age stratum.

### 3.3. Muscle Mass and Function

The results of muscle indexes are listed in Table 3. In detail, it took longer in the five-time chair stand test in the MPS group compared to the Control and MCI groups (*p* < 0.001); at the same time, MPS individuals presented weaker handgrip strength (*p* < 0.001). As for the 6-meter walk, the pace in the MCI group was remarkably slower than the Control group (*p* = 0.012), although no difference was observed between the MPS and the other two groups. Furthermore, body composition tests were conducted to evaluate the composition of muscle, fat, and so on in the body. Nevertheless, not only the muscle mass of limbs, but also upper arm circumference with or without fat did not vary significantly.

Stratified analyses were extended to physical performance measures, revealing sex- and age-specific patterns. Both the five-time chair stand test and handgrip strength demonstrated subgroup differences across gender and age strata that mirrored those observed in the general population. Notably, sex-specific disparities in gait speed between the Control and MCI groups emerged exclusively in male participants. Furthermore, age-stratified analyses identified distinctive patterns in older adults: within the ≥65 y subgroup, 6-meter walk in the MCI group was significantly reduced compared to both the Control and MPS groups, whereas such intergroup differences were not observed in younger participants.

### 3.4. Serum Biomarkers

To further verify the alterations of the serum biomarkers of cognition and muscle, the concentrations of Nfl, Aβ1-42, APP, BDNF, irisin, and CAF were conducted using an ELISA (*n* = 15~20/group), which is displayed in Figure 2. Notably, serum Nfl in the MCI (*p* = 0.005) and MPS (*p* = 0.001) groups significantly increased, and so did CAF (*p* < 0.05). On the contrary, a higher concentration of APP in the MPS group was detected in comparison to the Control (*p* = 0.002) and MCI (*p* = 0.040) groups.

### 3.5. Serum Oxysterols Concentrations

A total of 22 kinds of oxysterols were detected to evaluate their diversity of serum concentrations (Table 4). In particular, higher serum 27-OHC and 5α,6α-epoxy levels but lower 15-HC were presented in the MPS group (*p* < 0.05). Meanwhile, reduced 7k-25-OHC was observed in the Control group in contrast with the MCI (*p* = 0.016) and MPS groups (*p* = 0.006). In addition, 7α/β-OHC presented higher in the MPS group than the Control group (*p* = 0.002). The shift in serum oxysterols provides a possible connection between cognition and muscle decline.

### 3.6. Gut Microbiota

To investigate the alternation of gut microbiota in comorbidities, a 16S rDNA sequencing was conducted (*n* = 20/group). As depicted in the Venn diagram, the number of observed Amplicon Sequence Variants (ASVs) was reduced successively in the Control, MCI, and MPS groups (Figure 3A). The taxonomic composition of the community and relative abundance were remarkably different among groups (Figure 3B–F). In detail, MPS individuals presented a higher abundance of *Actinobacteriota* both at the phylum (*p* = 0.022) and class (*p* = 0.023) levels than the Control group (Figure 4A, B). In addition, the abundance of *Bifidobacteriales* in the MPS group at order, family, and genus levels (all *p* = 0.023) were enriched compared to the Control group (Figure 4C,D,G). As for *Bacteroidaceae*, there were decreased levels in the MPS group in comparison to the Control (*p* = 0.012) and MCI (*p* = 0.001) groups both at family and genus taxa (Figure 4D,E,G,H). In addition, *Rikenellaceae* and *Alistipes* revealed an enrichment in the Control group than the MCI (*p* = 0.050) and MPS groups (*p* = 0.021). Control subjects showed an elevated abundance of *Oscillospiraceae* (*p* = 0.013) at the family level, as well as *UCG-002* (*p* = 0.021), *UCG-005* (*p* = 0.040), and *[Ruminococcus]_torques_group* (*p* = 0.011) at genus-level than the MPS group (Figure 4D,G). When compared with the MCI group, the genus of *Coprococcus* (*p* = 0.021) and *Dorea* (*p* = 0.016) were enriched in the Control group (Figure 4I).

The analysis of α-diversity was evaluated using the indexes of Chao1, Observed_otus, Shannon, and Simpson, as shown in Figure 5A–D. Particularly, it was observed that Chao1 and Observed_otus in MCI and MPS were strikingly reduced than in the Control group, while the Shannon index was only significantly different between the Control and MPS group (*p* = 0.047). Based on the weighted UniFrac distance, both a principal co-ordinates analysis (PCOA) and non-metric multidimensional scaling (NMDS) analysis indicated that the species composition structure of the Control group was similar to the MCI group rather than the MPS group (Figure 5E,F). Furthermore, using LEfSe analysis, taxa *Akkermansia*, *Verrucomicrobiae*, and *Alistipes* were enriched in the Control group, while *Bacteroides* and *Lactobacillaceae* were enriched in the MCI group, along with *Bifidobacteriales*, *Actinobacteriota*, and *Megasphaera* in the MPS group, separately (Figure 6). The different colors of the cladogram represented the microbial taxa playing important roles in different groups (Figure 7).

### 3.7. Association of Cognition and Muscle Indexes

Spearman’s rank coefficients were calculated to estimate the correlation between cognition and muscle indicators based on the related data of all the three groups. Similarly to the heatmap shown in Figure 8, here we found that cognitive indicators were most significantly correlated with each other, and so were the muscle indexes. Furthermore, the five-time chair stand test was negatively correlated with a series of indexes including MMSE (r = −0.206; *p* < 0.001), MoCA (r = −0.302; *p* < 0.001), AVLT-SR (r = −0.179; *p* < 0.001), AVLT-LR (r = −0.182; *p* < 0.001), DSF (r = −0.158; *p* < 0.001), DSB (r = −0.148; *p* < 0.001), and SDMT (r = −0.312; *p* < 0.001), but positively correlated with TMT-A (r = 0.214; *p* < 0.001) and TMT-B (r = 0.217; *p* < 0.001). In addition, handgrip strength presented the same correlations with the five-time chair stand test except AVLT-SR and AVLT-LR, so did the 6-meter walk except AVLT-LR and DSTB (*p* < 0.05). As for the body composition scores, there was a positive association between the muscle mass of the right upper and left upper limbs as well as the limbs ASM with MMSE, MoCA, and DSF, while there was a negative association with TMT-A and TMT-B (*p* < 0.05). Furthermore, Nfl was negatively correlated with MoCA (r = −0.450; *p* < 0.001) and DSB (r = −0.275; *p* = 0.034), along with being positively correlated with calf circumference (r = 0.438; *p* < 0.001), the five-time chair stand test (r = 0.303; *p* = 0.019), and CAF (r = 0.576; *p* = 0.002). Similarly, CAF presented a negative relationship with MoCA (r = −0.396; *p* = 0.041), but a positive one with APP (r = 0.495; *p* = 0.027). Negative associations were observed between irisin and SCWT-RIE (r = −0.275; *p* = 0.046), but a positive association was observed with BDNF and the 6-meter walk (r = 0.327; *p* = 0.018), respectively.

### 3.8. Association of Oxysterols with Cognition Indexes

The correlation between oxysterols and cognition was analyzed as well (Figure 9). In detail, the global cognitive tests, MMSE and MoCA, were negatively associated with 7K-25-OHC and 5α,6α-epoxy (*p* < 0.05), while they were positively associated with 5α-6KC and C-4,6-dien-3 (*p* < 0.05). Moreover, MoCA was negatively associated with 4β-OHC (r = −0.201, *p* = 0.042) and 27-OHC (r = −0.220, *p* = 0.028) and positively associated with 15-HC (r = 0.251, *p* = 0.010). MMSE showed a positive association with C4 (r = 0.200, *p* = 0.043). As far as multidimensional cognitive tests were concerned, there was a negative correlation between DSF (r = −0.251, *p* = 0.011), SDMT (r = −0.249, *p* = 0.011) with 5α,6α-epoxy, TMT-B with 3β,5α,6β-triol (r = −0.213, *p* = 0.031), SCWT-TIE with 4α-OHC (r = −0.221, *p* = 0.025), 24S,25-epoxy (r = −0.196, *p* = 0.048) and C4 (r = −0.269, *p* = 0.006), as well as serum Nfl with 15-HC (r = −0.484, *p* < 0.001) and C-4,6-dien-3 (r = −0.290, *p* = 0.025). Meanwhile, there remained a positive correlation of AVLTSR with C4 (r = 0.212, *p* = 0.031), DSF with C-4,6-dien-3 (r = 0.207, *p* = 0.036), 4α-OHC with TMT-B (r = 0.223, *p* = 0.024), SCWT-TIE with 7-KC (r = 0.216, *p* = 0.028), DSB (r = 0.217, *p* = 0.028) and SCWT-RIE (r = 0.235, *p* = 0.017) with 15-HC, as well as serum Nfl with 7α/β-OHC (r = 0.292, *p* = 0.024), 7k-25-OHC (r = 0.356, *p* = 0.005) and 5α,6α-epoxy (r = 0.329, *p* = 0.010).

### 3.9. Association of Oxysterols with Muscle Indexes

To clarify the relationship between oxysterols and muscle indexes clearly, we conducted Spearman’s rank coefficients simultaneously (shown in Figure 10). In this case, calf circumference was negatively associated with 25-OHC (r = −0.236, *p* = 0.016) and 15-HC (r = −0.198, *p* = 0.045), and so was handgrip strength with 7α/β-OHC (r = −0.225, *p* = 0.023) and 7-KC (r = −0.201, *p* = 0.041), as well as five-time chair stand test with 15-HC (r = −0.219, *p* = 0.026), 3β,5α,6β-triol (r = −0.259, *p* = 0.008), and 3β,5α,6α-triol (r = −0.231, *p* = 0.019). On the contrary, calf circumference was proportional to 7k-25-OHC (r = 0.245, *p* = 0.013). Likewise, the five-time chair stand test revealed a positive relationship with 4α-OHC (r = 0.222, *p* = 0.024), 24S,25-epoxy (r = 0.197, *p* = 0.046) and 5α,6α-epoxy (r = 0.216, *p* = 0.029), while serum BDNF had positive correlation with 27-CA (r = 0.284, *p* = 0.034) and 7-HOCA (r = 0.277, *p* = 0.033).

## 4. Discussion

This study elucidated that global and multidimensional cognition were decreased in the MPS group successively when compared with the MCI group and Control group, along with worse performance in the five-time chair stand test and handgrip strength. In addition, 25-OHC decreased but 5α,6α-epoxy increased in the MPS group. Lower microbial diversity and shifted abundance including increased *Actinobacteriota* and *Bifidobacteriales* but reduced *Alistipes*, *Bacteroidaceae*, and *Rikenellaceae* were detected in the MPS group. Apart from that, there were significant correlations between serious cognitive indexes with muscle indexes, so was oxysterols with cognitive and muscle indicators. To our knowledge, our study is the first to explore the correlation between oxysterols and gut microbiota with the comorbidity of MCI and possible sarcopenia, which provides the foundation to reveal the pathogenesis of comorbidity subsequently.

As the most classic scales to evaluate global cognitive function, MMSE and MoCA were expansively applied to the diagnosis of MCI in epidemiologic studies, while the latter presented a higher ability to measure cognition [19]. Certainly, MoCA was conducted to preliminarily diagnose MCI in our research, which was worse in the Control, MCI, and MPS group successively, as was MMSE and multidimensional cognitive tests. Since the MCI and MPS group shared the same standard of cognition, the reduced score in the MPS group than the MCI group implied that the condition of possible sarcopenia may lead to cognitive impairment. Similarly, a cross-sectional study involving six low- and middle-income countries found that there existed a positive correlation between sarcopenia and MCI [20]. In addition, a longitudinal cohort in China reached the same conclusion that sarcopenia was associated with more severe cognitive impairment, including decreased cognition as well as performance in four dimensions of orientation, computation, memory, and drawing [2]. However, the cognitive tests used in these papers were different from ours, so further studies using the same tests should be considered. Likewise, with MoCA as the tool to evaluate cognitive function, community-dwelling subjects (over 80 years old) in China discovered a higher prevalence of MCI among sarcopenic elders, along with sarcopenia, low handgrip strength, and low gait speed as the independent risk factors of MCI [21]. Another study on participants in the age range of 60 years and above reached a similar conclusion [4], indicating that age range is not a restrictive factor affecting the correlation of comorbidity.

According to AWGS2019, grip strength and the five-time chair stand test are used to screen possible sarcopenia [17]. Handgrip strength is especially one of the main indexes used to assess muscle strength. In a Japanese cross-sectional study with the classification of possible sarcopenia by handgrip strength, sarcopenia was closely related to cognitive impairment in women, mainly due to the remarkable correlation between handgrip strength with immediate recall of logical memory and memory domain of ADAS-cog [22]. Nevertheless, another study used 26 kg for men and 16 kg for women as cut-off values, finding no potential relationship between handgrip strength with MCI patients progressed from cognitively normal individuals [23]. Therefore, it is suggested that the measurement standards should be unified among studies as far as possible, and the comparison of multiple indicators should be added to obtain more comparative conclusions. In addition to handgrip strength, gait speed is also one of the commonly used indicators. In Japanese diabetic subjects with MCI, although handgrip strength, gait speed, and bone mass index were decreased in MCI subjects, gait speed was the only decisive criterion for MCI [24]. Survival analysis showed that each 0.05 m/s slower gait speed was associated with a 7% increase in the risk of developing MCI/AD [25]. Based on the diagnostic criteria published by the European Working Group on Sarcopenia in Older People (EWGSOP2), grip strength and chair stand were recommended as indicators of muscle strength as well as gait speed for physical performance [26]. Several studies have used grip strength and gait speed to evaluate muscle function together [27,28]. Individuals with low handgrip strength and gait speed presented less likelihood of MCI, and one or more impaired sarcopenia components were related to limited cognitive function [29]. There are also studies that used the five-time chair stand test and gait speed to evaluate physical performance, while grip strength was used for muscle strength, which achieved similar correlation results with the above studies [2]. Our findings showed that the MPS group had a significantly longer duration of the five-time chair stand test and worse handgrip strength than Control and MCI groups. However, there was no significant difference in gait speed of the 6-meter walk in MPS group. In particular, these indexes were positively correlated with multiple cognitive indicators, such as MMSE, MOCA, DSF, and SDMT, which revealed the representative role of cognitive and muscle metrics in linking muscle–brain function.

In addition to anthropometric data, our study also measures the concentration of serum biomarkers of cognition and muscle. The results exhibit that Nfl increased significantly in the MCI and MPS groups. Nfl was a putative biomarker for cognitive dysfunction, which was detected with a higher level in the peripheral circulation of cognitively impaired individuals, and so did Aβ1-42 and APP [30]. A higher concentration of Nfl was detected in MCI Aβ+ and AD Aβ+ individuals and is associated with prospective cognitive decline [31]. Notably, Nfl has also been found to be memorably elevated in sarcopenia subjects and negatively correlated with handgrip strength [32]. In our study, Nfl was not only higher in the MPS group than the Control group, but also is significantly positively associated with the duration of the five-time chair stand test and CAF level, which suggested the potential of Nfl as a co-biomarker of cognition and muscle. Besides, it has also been found that, as the precursor of Aβ, APP’s high expression in skeletal muscle led to sarcopenia-like defects in AD mice, indicating the role of the axis of muscular APP to the brain in the development of AD [33]. Our study shows that APP concentration was significantly higher in the MPS group than the Control and MCI groups, which again proved the dual role of APP in cognitive and muscle dysfunction. Elevated peripheral CAF levels, an early biomarker of sarcopenia, were also found to be elevated in patients with different stages of AD, and higher levels were detected in more severe stages of AD [34], which is consistent with the results of our study. Irisin, a myokine to prevent muscle atrophy, has been found to be evidently reduced in the serum of sarcopenia patients [35]. However, another cross-sectional study found that serum irisin levels presented no relationship with sarcopenia status and relevant clinical muscle parameters, which did not support the conclusion that Irisin can be used as a biomarker of sarcopenia in the population-based study [36]. In our study, serum irisin levels were not significantly varied among groups in our study, but were negatively correlated with the score of SCWT-RIE, again not supporting the prediction of sarcopenia by reducing irisin. These findings suggest that more studies are needed to identify whether irisin is a qualified biomarker of sarcopenia.

In addition to the aforementioned assessment of serum biomarkers and muscle strength, muscle mass was also quantified. However, no intergroup differences were observed in appendicular muscle mass, suggesting that muscle functional decline may stem from impaired neural-to-muscle communication rather than muscle mass deterioration. The neural–muscular communication system is mediated by the neuromuscular junction (NMJ), a highly specialized chemical synapse that converts presynaptic motor neuron action potentials into postsynaptic muscle fiber contractions [37]. Age-related NMJ remodeling involves proteolytic cleavage of Agrin into CAF, which inactivates cholinergic receptors in presynaptic membranes and promotes denervated muscle fiber accumulation—a hallmark feature of sarcopenia. This mechanistic pathway supports serum CAF as a potential early diagnostic biomarker for sarcopenia [38]. Meta-analytical evidence demonstrates significant correlations between CAF levels and sarcopenia-related parameters including appendicular lean mass, handgrip strength, and gait velocity [39]. The present study reveals significantly elevated CAF levels in both MCI and MPS groups, indicating substantial NMJ dysfunction in populations with cognitive impairment and muscular deterioration, which may underlie observed alterations in muscle performance. These findings underscore the importance of targeting NMJ structural integrity and functional capacity as critical therapeutic strategies for sarcopenia prevention and management.

Oxysterols have attracted increasing attention due to their role in neurodegenerative diseases. Among them, 27-OHC, as the most abundant oxysterol in peripheral circulation, can freely penetrate the BBB and play neurotoxic effects through a series of mechanisms including the promotion of Aβ deposition, Th17/Treg-related neuroimmune response, and abnormal glucose uptake, which have been well described in our previous studies. Population studies have also shown that elevated serum 27-OHC level is an independent risk factor for MCI [11]. In contrast, 24S-OHC exerted neuroprotective effects against 27-OHC-induced cognitive impairment through RORγt-related immune responses [40]. 25-OHC is another immune-related side-chain oxysterol, which promoted AD-related neuroinflammation as a mediator secreted by microglia [41]. In addition, 5α,6α-epoxy, a derivative of cholesterol autoxidation, was dramatically increased in the brain tissue of advanced AD patients [42]. Nevertheless, no oxysterol has been found to play a part in sarcopenia, except for 7β-OHC and 7-KC. Higher concentrations of 7β-OHC and 7-KC were detected in the plasma of sarcopenic patients, along with enhanced oxidative stress and myocyte toxicity [12]. Our results showed higher 27-OHC and 7α/β-OHC in the MPS group. Moreover, the concentration of 7K-25-OHC in the MPS and MCI groups was remarkable elevated. Meanwhile, all of the above oxysterols are observed to be associated with cognition function indexes. In addition, 7α/β-OHC, and 7-KC were negatively correlated with handgrip strength, while 7K-25-OHC was positively correlated with calf circumference. Thus, it can be seen oxysterols are closely related to cognition and muscle function. It is worth mentioning that 5α,6α-epoxy in the MPS group was higher than the Control group, and displayed negative correlations with MMSE, MoCA, DSF, and SDMT, but positive correlations with Nfl and the duration of the five-time chair stand test. 5α,6α-epoxy exhibited higher quality to evaluate comorbidity since its high levels were associated with worse cognitive and muscle function. Therefore, aberrantly shifted oxysterols have potential as comorbid biomarkers of cognitive decline and muscular dystrophy.

With the gut–brain axis and gut–muscle axis having gradually become part of the widely accepted consensus, the role of gut microbiota in the co-dysfunction of cognition and muscle deserves further study [43]. However, most of the current research papers mainly focus on the alterations of gut microbiota in either MCI/AD or sarcopenia solely. To the best of our knowledge, our study is the first to focus on the abnormalities of gut microbiota in comorbidities. Reduced microbiota diversity and decreased abundance of specific microbes have been demonstrated in MCI/AD as well as sarcopenia [44,45]. Similarly, our results show that Chao1 and Observed_otus in the MPS group and MCI group were dramatically lower than the Control group, and the MPS group presented a downward trend compared to the MCI group. In addition, the abundance of microorganisms also changed between groups. Similarly, our results show that Chao1 and Observed_otus were significantly lower in the MPS and MCI groups than in the Control group, and tended to decrease in the MPS group compared to the MCI group. In addition, the abundance of microbiota was also altered between groups. Among them, the abundance of *Alistipes* and *Rikenellaceae* decreased considerably in the MPS and MCI group compared to the Control group, and the alteration of the former was also confirmed in the previous study of our team [46]. It was demonstrated that *Alistipes* was obviously consumed in the microbiota of sarcopenia patients [47], which hinted that *Alistipes* may be one of the key strains connecting the gut–brain–muscle axis. Of course, a variety of gut microbiota with altered abundance are worthy of further exploration, regarding the possible mechanisms in comorbidity, to bring a little bit of potential brightness to the black box of gut microbiota mechanisms in the comorbidity of AD and sarcopenia.

This research has several limitations that should be acknowledged. Firstly, owing to cost constraints, not all samples underwent oxysterol and biomarker analyses, which may limit the representativeness of these biochemical measures compared to the more comprehensively assessed cognitive and muscular outcomes. Future large-scale investigations are warranted to validate the reliability of these findings. Secondly, the cohort was exclusively derived from a Chinese population, potentially restricting the generalizability of the observed cognitive-muscular dysfunction relationships to global populations with distinct genetic and environmental backgrounds. Multi-center cohorts involving diverse ethnicities are critical to elucidate the universality and mechanisms of cognitive–muscular comorbidities. Finally, as an observational study, our findings primarily establish associations between oxysterols, gut microbiota, and cognitive–muscular comorbidities. However, whether oxysterols mechanistically influence these comorbidities through gut microbial modulation remains speculative and requires validation via targeted mechanistic animal studies.

## 5. Conclusions

This study confirms the abnormal serum oxysterols and gut microbiota characteristics of MCI individuals with or without possible sarcopenia. Shifted oxysterols and gut microbiota may be the crucial core of comorbid cognitive impairment and muscle atrophy, which deserves further exploration. In the future, priority should be given to elucidating causal pathways (e.g., gut–brain–muscle axis via animal models), integrating multi-omics (metabolomics/metagenomics) to identify biomarkers, and testing interventions (probiotics/diet) to restore homeostasis. Expanding into diverse global cohorts will clarify the universality of these findings, while translating discoveries into dual-purpose diagnostic/therapeutic strategies could mitigate aging-related brain and muscle decline.

## Figures and Tables

**Figure 1 nutrients-17-01277-f001:**
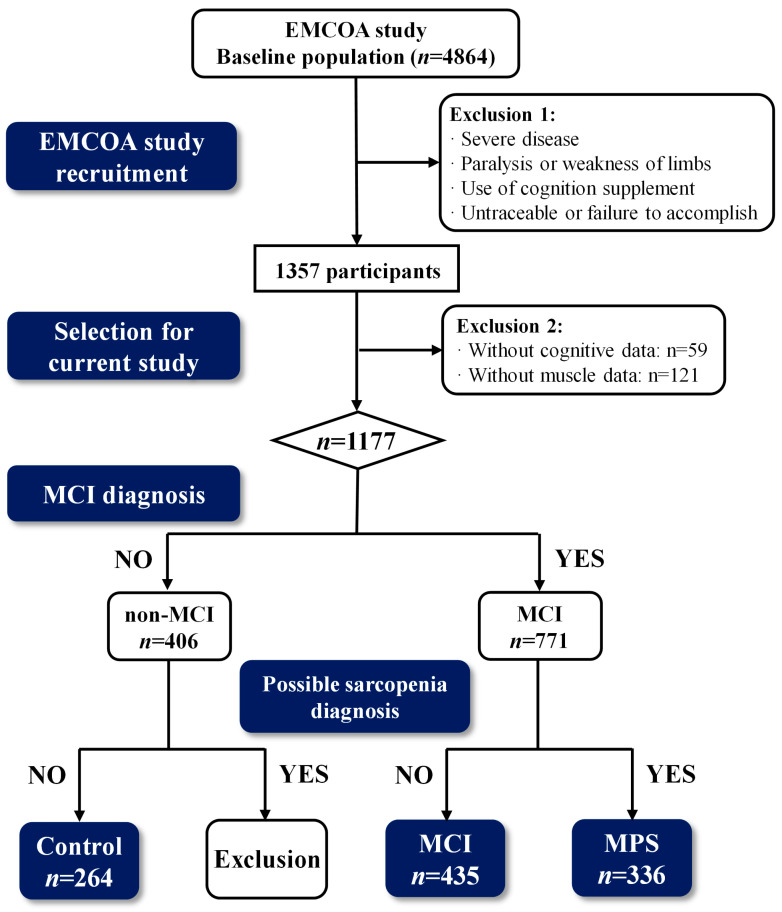
Flow chart of detailed recruitment in the study. MCI, mild cognitive impairment; MPS, MCI with possible sarcopenia.

**Figure 2 nutrients-17-01277-f002:**
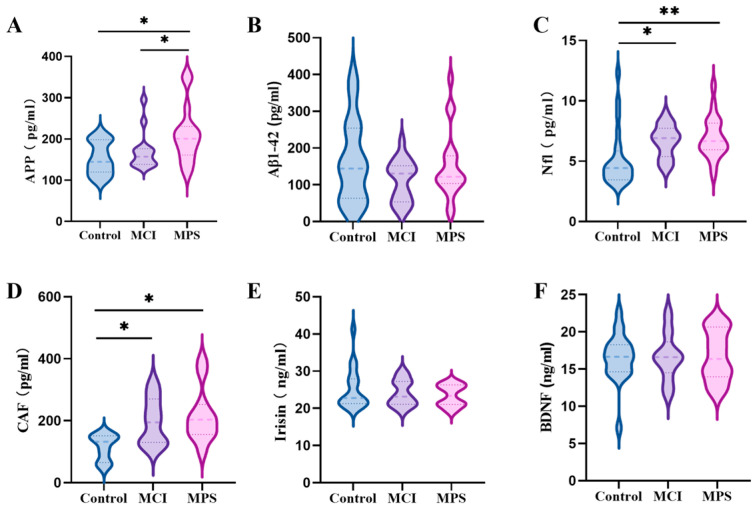
Serum biomarkers detected using ELISA. (**A**) Amyloid precursor protein (APP), (**B**) amyloid-β (Aβ1-42), (**C**) neurofilament (Nfl), (**D**) C-terminal agrin fragment (CAF), (**E**) irisin, and (**F**) brain-derived neurotrophic factor (BDNF). * *p* < 0.05, ** *p* < 0.01.

**Figure 3 nutrients-17-01277-f003:**
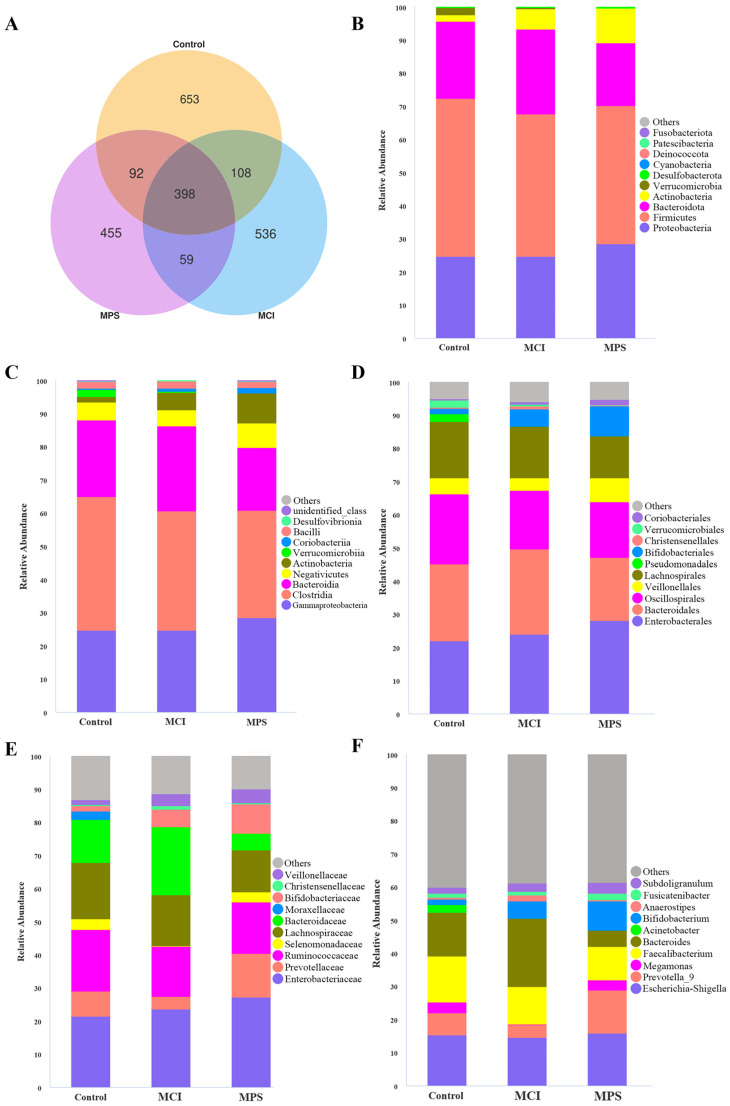
Gut microbiota composition among Control, MCI, and MPS group. (**A**) Venn diagram illustrating the amount of ASVs. Relative abundance of top ten taxa on the level of (**B**) phylum, (**C**) class, (**D**) order, (**E**) family, and (**F**) genus.

**Figure 4 nutrients-17-01277-f004:**
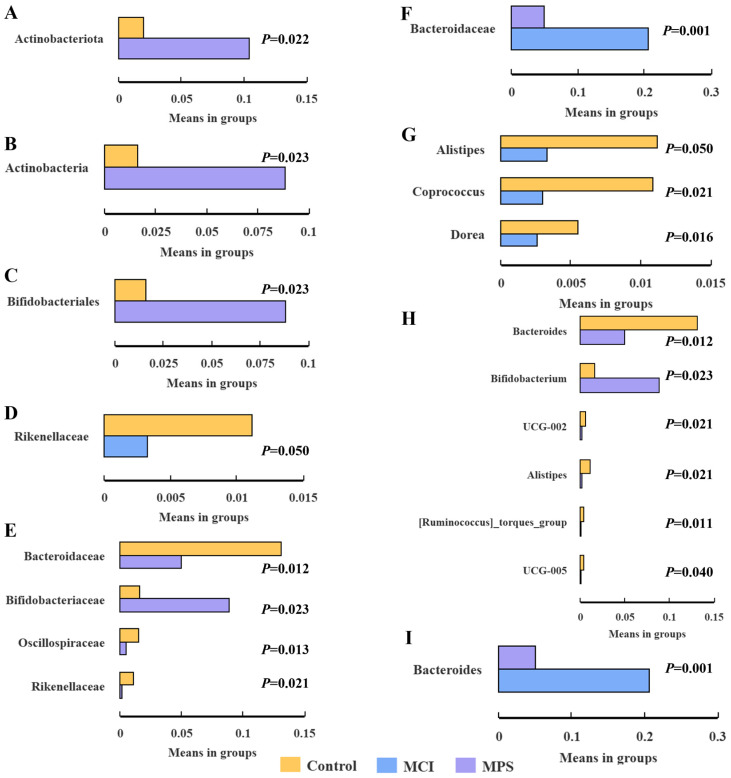
Different relative abundance of bacterial taxa using *t*-test. Taxa were presented in the level of (**A**) phylum, (**B**) class, (**C**) order, (**D**) family (Control vs. MPS), (**E**) family (MCI vs. MPS), (**F**) family (Control vs. MCI), (**G**) genus (Control vs. MPS), (**H**) genus (MCI vs. MPS), and (**I**) genus (Control vs. MCI). *p* < 0.05 were considered significant.

**Figure 5 nutrients-17-01277-f005:**
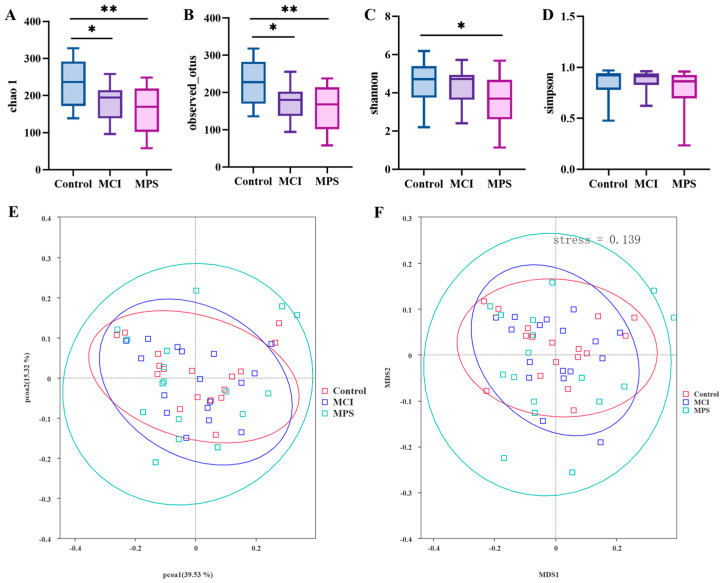
Microbial diversity and cladogram. The α-diversity of the fecal microbiome including (**A**) Chao 1, (**B**) Observed_otus, (**C**) Shannon, and (**D**) Simpson was analyzed using the Wilcoxon rank-sum test. The β-diversity was analyzed using PCOA (**E**) and NMDS (**F**) on the basis of weighted UniFrac distance. * *p* < 0.05, ** *p* < 0.05.

**Figure 6 nutrients-17-01277-f006:**
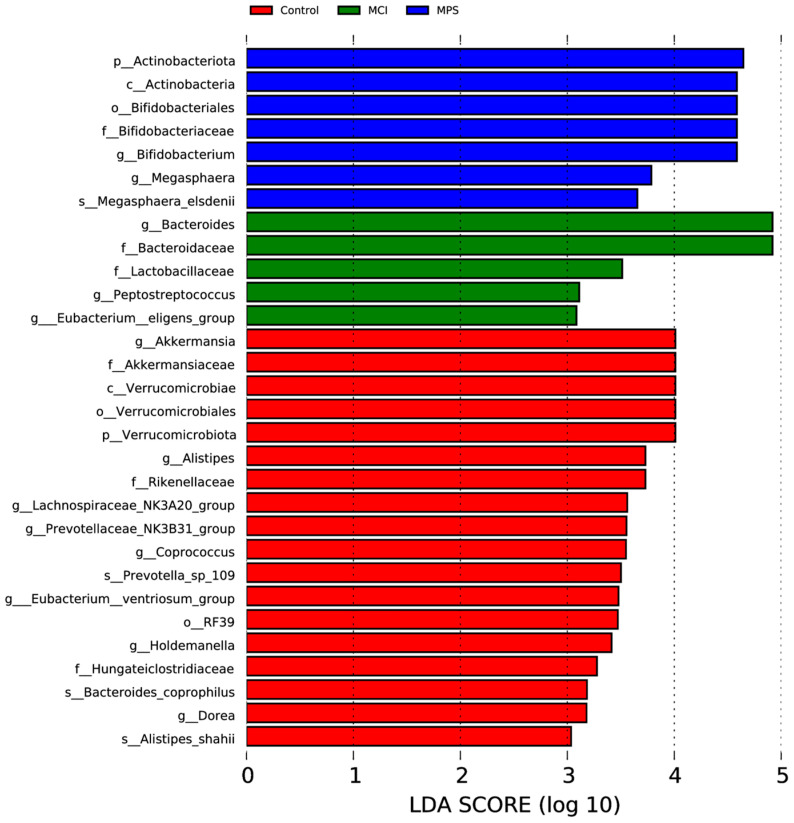
Linear discriminant analysis (LDA) effect size (LEfSe) analysis classifying the taxonomic differences among groups. The cutoff value of the LDA was 3.0, with *p* < 0.05 showing significant values.

**Figure 7 nutrients-17-01277-f007:**
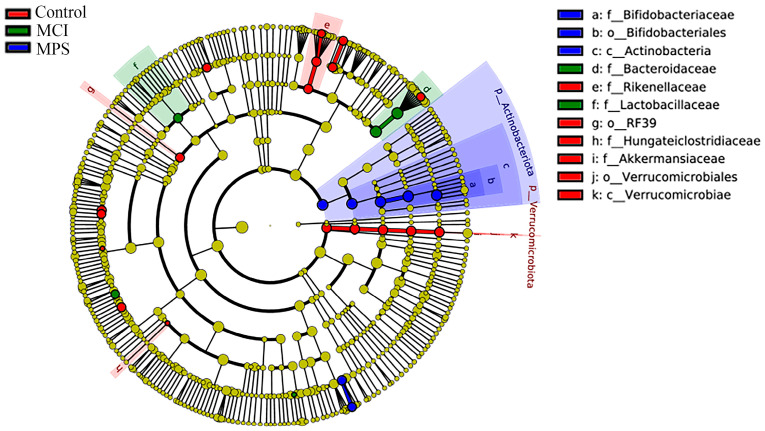
Cladogram of bacterial signatures based on differential gut microbial taxa from LEfSe analysis.

**Figure 8 nutrients-17-01277-f008:**
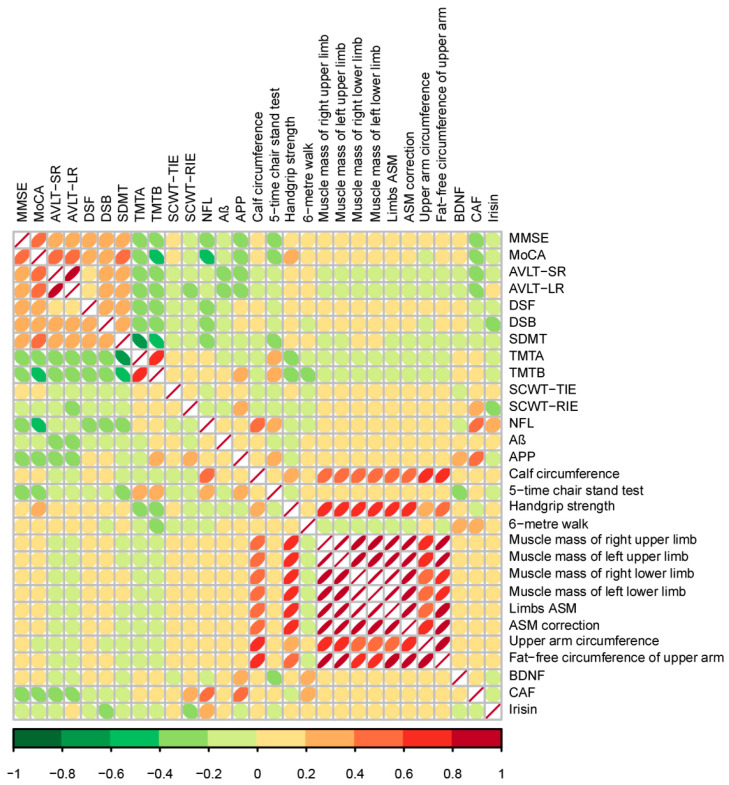
Correlation between the indexes of cognition and muscle using Spearman’s correlation coefficient. The strength of association is represented by color.

**Figure 9 nutrients-17-01277-f009:**
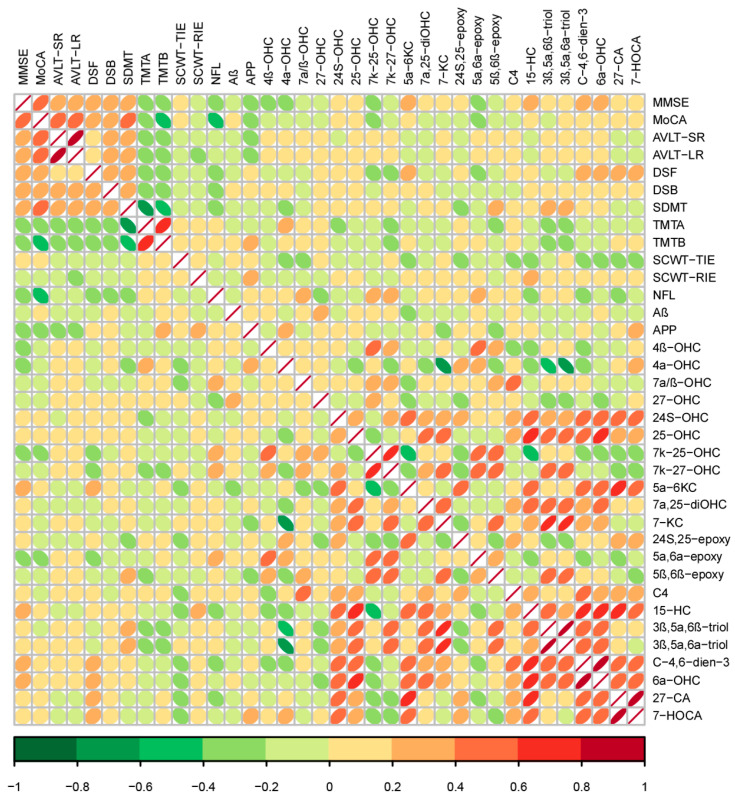
Correlation between oxysterols and cognition indexes using Spearman’s correlation coefficient. The strength of association is represented by color.

**Figure 10 nutrients-17-01277-f010:**
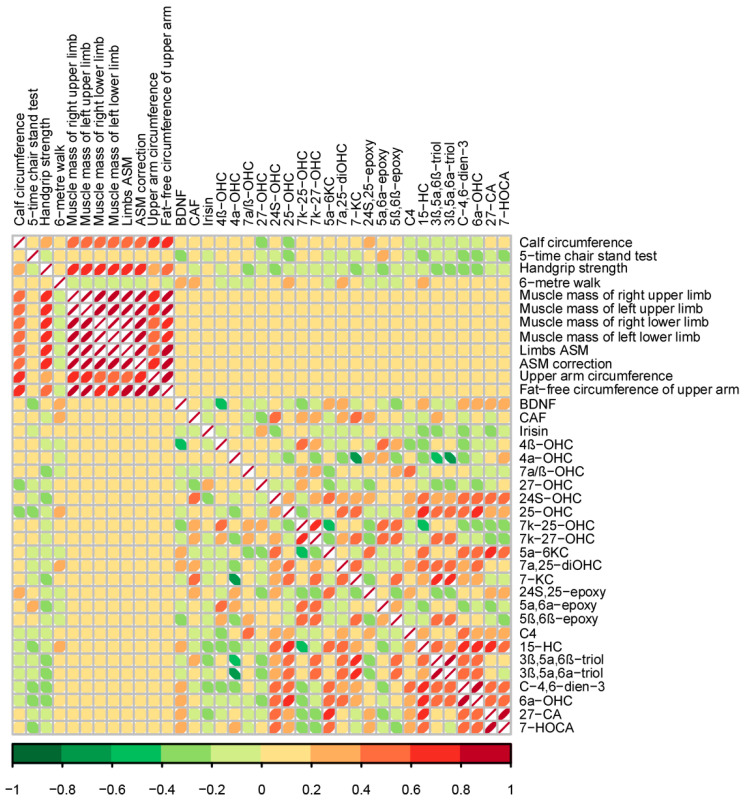
Correlation between oxysterols and muscle indexes using Spearman’s correlation coefficient. The strength of association is represented by color.

**Table 1 nutrients-17-01277-t001:** Demographic characteristics, serum lipids, and dietary nutrient intake.

Index	Control	MCI	MPS	*p*
General characteristics (*n* = 1035)				
Male, *n* (%)	105 (39.7)	170 (39.1)	141 (42.0)	0.711
Age (years)	62 ± 7	61.5 ± 8	65 ± 8	<0.001 *
Education year (*n*)				<0.001 *
<9	136	254	228	
9~12	93	147	91	
>12	33	34	17	
Height (cm)	163.00 ± 11.00	163.00 ± 11.00	162.00 ± 11.00	0.035 *
Weight (kg)	64.40 ± 12.30	64.70 ± 11.80	63.80 ± 15.10	0.592
BMI (kg/m^2^)	24.10 ± 0.16	24.26 ± 0.14	24.42 ± 0.17	0.412
Smoker, *n* (%)	46 (18.6)	87 (20.4)	63 (19.1)	0.825
Drinker, *n* (%)	28 (11.3)	56 (13.1)	51 (15.5)	0.346
Serum lipids (*n* = 1032)				
TC (mmol/L)	4.87 ± 1.42	4.82 ± 1.50	4.80 ± 1.50	0.282
TG (mmol/L)	1.48 ± 0.89	1.58 ± 0.94	1.49 ± 0.91	0.258
HDL-C (mmol/L)	1.37 ± 0.41	1.34 ± 0.41	1.28 ± 0.45	0.001 *
LDL-C (mmol/L)	2.57 ± 0.99	2.58 ± 1.16	2.57 ± 1.18	0.522
Dietary nutrients intake (*n* = 957)				
Energy (kcal/day)	1623.72 ± 702.25	1577.89 ± 773.45	1597.46 ± 777.08	0.477
Protein (g/day)	61.03 ± 36.02	59.40 ± 35.45	60.55 ± 34.59	0.638
Fat (g/day)	70.54 ± 43.63	68.70 ± 47.06	70.06 ± 44.06	0.844
Cholesterol (mg/day)	439.50 ± 226.35	448.50 ± 183.19	453.50 ± 114.55	0.296

Notes: MCI, mild cognitive impairment; MPS, MCI with possible sarcopenia; BMI, body mass index; TC, total cholesterol; TG, triglycerides; HDL-C, high-density lipoprotein cholesterol; LDL-C, low-density lipoprotein cholesterol. * *p* < 0.05.

**Table 2 nutrients-17-01277-t002:** Cognitive test scores.

Index	Control	MCI	MPS	*p*
Global cognition (*n* = 1035)				
MMSE	29 ± 2	27 ± 3	27 ± 3	<0.001 *
Male	29 ± 2	28 ± 2	27 ± 2	<0.001 *
Famale	29 ± 3	28 ± 2	28 ± 2	<0.001 *
<65 y	29 ± 1	28 ± 2	28 ± 2	<0.001 *
≥65 y	29 ± 2	29 ± 2	27 ± 1	<0.001 *
MoCA	25 ± 2	22 ± 3	21 ± 4	<0.001 *
Male	25 ± 2	23 ± 2.5	23 ± 3	<0.001 *
Famale	25 ± 2	23 ± 3	22 ± 3	<0.001 *
<65 y	25 ± 2	23 ± 3	22 ± 3.5	<0.001 *
≥65 y	26 ± 2	23 ± 3	23 ± 2	<0.001 *
Multidimensional cognition (*n* = 1002)				
AVLT-SR	6 ± 4	5 ± 4	5 ± 3	<0.001 *
Male	5.5 ± 3	5 ± 3	4 ± 2	<0.001 *
Famale	6 ± 3	5 ± 3.75	6 ± 3	<0.001 *
<65 y	6 ± 3	5 ± 4	5 ± 3	<0.001 *
≥65 y	6 ± 3	4 ± 5	4 ± 3	<0.001 *
AVLT-LR	5 ± 4	4 ± 4	4 ± 3	<0.001 *
Male	4 ± 3.75	4 ± 3	3 ± 3	<0.001 *
Famale	6 ± 4	4 ± 3	4 ± 4	<0.001 *
<65 y	5 ± 4	4 ± 3	3.5 ± 4.75	<0.001 *
≥65 y	5.5 ± 3	3 ± 4	4 ± 3	<0.001 *
SDMT	36 ± 17	35 ± 13.5	31 ± 12	<0.001 *
Male	36 ± 13	35 ± 13	30.5 ± 10	<0.001 *
Famale	38 ± 20	35 ± 14.75	32 ± 16	<0.001 *
<65 y	37 ± 15	36 ± 14.25	31 ± 14	<0.001 *
≥65 y	35 ± 19.5	31.5 ± 8.75	31 ± 12	<0.001 *
DSF	8 ± 2	7 ± 1	8 ± 1	<0.001 *
Male	8 ± 2	8 ± 1	8 ± 1	0.001 *
Famale	8 ± 2	7 ± 2	8 ± 2	<0.001 *
<65 y	8 ± 2	7 ± 1	8 ± 2	<0.001 *
≥65 y	8 ± 1	7 ± 1.25	8 ± 1	0.015 *
DSB	4 ± 1	4 ± 1	4 ± 1	<0.001 *
Male	4 ± 2	4 ± 2	4 ± 1	0.004*
Famale	4 ± 1	4 ± 1	4 ± 1	<0.001 *
<65 y	4 ± 1	4 ± 1.25	4 ± 1	<0.001 *
≥65 y	4 ± 2	3 ± 1	4 ± 1	<0.001 *
TMT-A	60 ± 30	62 ± 35	70 ± 39	<0.001 *
Male	57 ± 18.25	59 ± 26.5	67 ± 28	<0.001 *
Famale	62 ± 34	64.5 ± 38	76 ± 39	<0.001 *
<65 y	60 ± 29	62 ± 34.75	70.5 ± 39	<0.001 *
≥65 y	57.5 ± 37.5	66.5 ± 25.75	68 ± 35	<0.001 *
TMT-B	144 ± 72	161 ± 90	186 ± 94	<0.001 *
Male	142 ± 60.25	156 ± 88.5	175.5 ± 90.75	<0.001 *
Famale	147 ± 75	170 ± 99.75	190 ± 103	<0.001 *
<65 y	140 ± 72	160 ± 90.25	190.5 ± 93	<0.001 *
≥65 y	149 ± 75.75	170.5 ± 108.25	160 ± 83	<0.001 *
SCWT-TIE	36 ± 13	35 ± 18.5	36 ± 18	0.774
Male	40 ± 18	42 ± 23.5	43.5 ± 21.5	0.370
Famale	34 ± 13	33 ± 15	32 ± 16	0.539
<65 y	35 ± 13	34 ± 17	35 ± 18	0.807
≥65 y	39.5 ± 18	39.5 ± 19	42 ± 22	0.723
SCWT-RIE	0 ± 2	0 ± 2	0 ± 2	0.848
Male	0 ± 2	0 ± 2	0 ± 3.75	0.508
Famale	0 ± 2	0 ± 3	0 ± 2	0.351
<65 y	0 ± 2	0 ± 2	0 ± 2.75	0.932
≥65 y	0 ± 1.75	1 ± 4	0 ± 1	0.158

Notes: MCI, mild cognitive impairment; MPS, MCI with possible sarcopenia; MMSE, mini-mental state examination; MoCA, Montreal cognitive assessment; AVLT-SR, auditory verbal learning test-short recall; AVLT-LR, auditory verbal learning test-long recall; SDMT, symbol digit modalities test; DSF, digit span forwards; DSB, digit span backwards; TMT-A, trail making test-A; TMT-B, trail making test-B; SCWT-TIE, Stroop color word test-time interfered effects; SCWT-RIE, Stroop color word test-reaction interfered effects. * *p* < 0.05.

**Table 3 nutrients-17-01277-t003:** Muscle mass and function.

Index	Control	MCI	MPS	*p*
Muscle function (*n* = 1023)				
Five-time chair stand test (s)	9.31 ± 2.43	9.70 ± 2.55	13.24 ± 2.77	<0.001 *
Male	8.99 ± 2.27	9.61 ± 2.36	13.59 ± 2.84	<0.001 *
Famale	9.40 ± 2.55	9.75 ± 2.66	12.98 ± 2.79	<0.001 *
<65 y	9.24 ± 2.32	9.59 ± 2.58	12.75 ± 2.80	<0.001 *
≥65 y	9.35 ± 2.44	9.80 ± 2.42	13.52 ± 2.99	<0.001 *
Handgrip strength (kg)	26.10 ± 13.00	25.60 ± 11.43	22.40 ± 10.35	<0.001 *
Male	37.00 ± 8.60	35.35 ± 8.28	31.10 ± 7.15	<0.001 *
Famale	22.95 ± 5.40	22.70 ± 4.65	18.90 ± 6.50	<0.001 *
<65 y	25.50 ± 9.27	25.40 ± 11.00	22.50 ± 9.70	<0.001 *
≥65 y	29.50 ± 13.40	26.90 ± 12.10	22.30 ± 11.40	<0.001 *
6-meter walk (m/s)	1.13 ± 0.44	1.02 ± 0.45	1.01 ± 0.30	0.013 *
Male	0.83 ± 0.21	0.82 ± 0.16	0.94 ± 0.20	0.031 *
Famale	0.85 ± 0.23	0.84 ± 0.23	0.95 ± 0.25	0.306
<65 y	0.85 ± 0.24	0.83 ± 0.15	0.89 ± 0.19	0.056
≥65 y	0.83 ± 0.19	0.83 ± 0.22	0.97 ± 0.27	<0.001 *
Muscle mass (*n* = 528)				
Waistline (cm)	85.00 ± 10.50	84.00 ± 11.13	86.00 ± 12.55	0.347
Male	90.10 ± 7.81	88.08 ± 7.88	88.39 ± 7.65	0.776
Famale	83.00 ± 10.13	82.00 ± 10.50	84.00 ± 11.00	0.256
<65 y	83.50 ± 13.55	83.00 ± 11.00	84.00 ± 13.00	0.631
≥65 y	85.50 ± 10.00	85.00 ± 11.00	87.00 ± 11.50	0.306
Hipline (cm)	97.00 ± 8.75	96.00 ± 7.00	97.00 ± 8.00	0.488
Male	101.00 ± 6.60	98.00 ± 5.80	98.00 ± 8.00	0.330
Famale	96.40 ± 7.78	95.00 ± 7.13	96.00 ± 8.00	0.569
<65 y	97.00 ± 9.00	96.00 ± 5.50	96.00 ± 9.00	0.644
≥65 y	97.50 ± 7.50	96.00 ± 8.00	97.00 ± 7.00	0.443
Calf circumference (cm)	34.50 ± 4.10	34.00 ± 3.25	34.00 ± 3.50	0.425
Male	36.00 ± 4.30	35.75 ± 4.88	35.50 ± 3.50	0.440
Famale	33.50 ± 4.08	33.50 ± 3.00	33.00 ± 3.00	0.505
<65 y	34.25 ± 4.35	34.00 ± 3.00	34.00 ± 3.50	0.194
≥65 y	34.50 ± 3.80	34.00 ± 4.00	34.00 ± 4.00	0.663
Muscle mass of right upper limb (kg)	2.14 ± 0.92	2.16 ± 0.79	2.18 ± 0.93	0.557
Male	2.93 ± 0.71	2.90 ± 0.59	2.91 ± 0.56	0.839
Famale	2.01 ± 0.35	2.00 ± 0.31	1.92 ± 0.46	0.079
<65 y	2.09 ± 0.64	2.13 ± 0.70	2.04 ± 0.76	0.150
≥65 y	2.38 ± 0.96	2.31 ± 0.86	2.25 ± 0.93	0.817
Muscle mass of left upper limb (kg)	2.12 ± 0.89	2.12 ± 0.76	2.12 ± 0.90	0.658
Male	2.93 ± 0.43	2.87 ± 0.40	2.84 ± 0.46	0.851
Famale	1.97 ± 0.38	1.97 ± 0.28	1.88 ± 0.44	0.120
<65 y	2.06 ± 0.65	2.10 ± 0.67	1.99 ± 0.70	0.146
≥65 y	2.35 ± 0.84	2.23 ± 0.79	2.23 ± 0.98	0.830
Muscle mass of right lower limb (kg)	6.37 ± 2.15	6.44 ± 1.96	6.37 ± 2.35	0.491
Male	8.39 ± 1.37	8.07 ± 1.24	8.10 ± 1.32	0.354
Famale	6.05 ± 0.83	5.97 ± 0.96	5.67 ± 1.03	0.066
<65 y	6.22 ± 1.35	6.34 ± 1.67	6.05 ± 1.98	0.235
≥65 y	7.05 ± 2.66	6.66 ± 2.08	6.55 ± 2.66	0.818
Muscle mass of left lower limb (kg)	6.29 ± 2.10	6.43 ± 2.02	6.38 ± 2.24	0.465
Male	8.25 ± 1.55	8.04 ± 1.24	8.03 ± 1.32	0.412
Famale	5.99 ± 0.78	5.93 ± 0.95	5.70 ± 1.04	0.061
<65 y	6.22 ± 1.48	6.32 ± 1.67	6.06 ± 1.75	0.210
≥65 y	7.11 ± 2.38	6.70 ± 2.20	6.57 ± 2.50	0.830
Limb ASM	17.08 ± 5.78	17.10 ± 5.51	17.11 ± 6.58	0.512
Male	22.49 ± 3.71	21.87 ± 3.77	21.95 ± 3.49	0.542
Famale	15.74 ± 2.44	16.01 ± 2.49	15.07 ± 2.89	0.054
<65 y	16.66 ± 3.85	16.86 ± 4.41	16.08 ± 5.48	0.199
≥65 y	18.77 ± 6.75	18.03 ± 6.05	17.43 ± 7.11	0.848
ASM correction	6.62 ± 1.44	6.64 ± 1.22	6.63 ± 1.48	0.679
Male	7.65 ± 0.83	7.51 ± 0.89	7.69 ± 0.96	0.900
Famale	6.23 ± 0.80	6.31 ± 0.66	6.11 ± 0.76	0.081
<65 y	6.40 ± 1.14	6.51 ± 1.19	6.40 ± 1.18	0.190
≥65 y	6.98 ± 1.43	6.75 ± 1.30	6.70 ± 1.57	0.758
Upper arm circumference (cm)	30.60 ± 4.00	30.40 ± 3.50	30.40 ± 3.63	0.801
Male	31.50 ± 2.30	31.45 ± 3.52	31.60 ± 3.60	0.940
Famale	29.90 ± 3.60	29.60 ± 2.80	29.90 ± 3.50	0.936
<65 y	30.25 ± 4.55	30.50 ± 3.00	29.90 ± 3.75	0.673
≥65 y	31.00 ± 2.90	30.40 ± 3.70	30.90 ± 3.60	0.348
Fat-free circumference of upper arm (cm)	24.10 ± 3.00	23.90 ± 2.73	24.10 ± 3.25	0.954
Male	26.30 ± 1.20	26.00 ± 2.50	26.20 ± 2.45	0.966
Famale	23.20 ± 1.85	23.30 ± 1.60	23.30 ± 2.10	0.831
<65 y	23.40 ± 2.90	23.90 ± 2.30	23.80 ± 2.85	0.306
≥65 y	24.60 ± 2.70	24.00 ± 3.00	24.30 ± 3.10	0.565

Notes: MCI, mild cognitive impairment; MPS, MCI with possible sarcopenia; ASM, appendicular lean mass. * *p* < 0.05.

**Table 4 nutrients-17-01277-t004:** Serum levels of oxysterols.

Index (μmol/L)	Control (*n* = 29)	MCI (*n* = 47)	MPS (*n* = 27)	*p*
27-OHC	0.0464 (0.0346,0.0816)	0.0492 (0.0324,0.0700)	0.0728 (0.0503,0.0789)	0.033 *
4β-OHC	0.6369 (0.4988,1.2422)	0.8150 (0.6061,1.1742)	1.0774 (0.6927,1.4093)	0.107
4α-OHC	0.1223 (0.0786,0.2955)	0.1407 (0.0949,0.2788)	0.1540 (0.0961,0.2646)	0.793
7α/β-OHC	0.0061 (0.0042,0.0084)	0.0070 (0.0054,0.0095)	0.0094 (0.0059,0.0148)	0.009 *
24S-OHC	0.6225 (0.4836,0.7844)	0.6073 (0.4564,0.74770)	0.5640 (0.4178,0.7263)	0.643
25-OHC	0.0011 (0.0007,0.0034)	0.0018 (0.0007,0.0097)	0.0009 (0.0006,0.0026)	0.155
7k-25-OHC	0.0376 (0.0251,0.0744)	0.0637 (0.0386,0.1033)	0.0704 (0.0467,0.1098)	0.012 *
7k-27-OHC	0.0457 (0.0259,0.0632)	0.0529 (0.0353,0.0769)	0.0595 (0.0515,0.0842)	0.286
5α-6KC	0.0109 (0.0072,2.6184)	0.0103 (0.0051,0.0161)	0.0083 (0.0046,0.0131)	0.082
7α,25-diOHC	0.0105 (0.0064,0.0210)	0.0153 (0.0101,0.0219)	0.0126 (0.0076,0.0205)	0.084
7-KC	0.0657 (0.0316,0.2580)	0.1099 (0.0530,0.2571)	0.1447 (0.0718,0.2100)	0.235
24S,25-epoxy	0.0726 (0.0567,0.2303)	0.1027 (0.0586,0.1546)	0.1166 (0.0732,0.1452)	0.575
5α,6α-epoxy	0.0491 (0.0310,0.0723)	0.0571 (0.0374,0.0756)	0.0763 (0.0615,0.1142)	0.001 *
5β,6β-epoxy	0.1518 (0.1111,0.2004)	0.1686 (0.0961,0.2630)	0.2174 (0.1215,0.2963)	0.105
C4	0.0363 (0.0286,0.0633)	0.0530 (0.0312,0.0838)	0.0423 (0.0232,0.0678)	0.342
15-HC	0.0477 (0.0211,0.0671)	0.0382 (0.0211,0.0702)	0.0250 (0.0148,0.0428)	0.048 *
3β,5α,6β-triol	0.0967 (0.0087,0.2582)	0.1145 (0.0501,0.3500)	0.1099 (0.0689,0.1946)	0.466
3β,5α,6α-triol	0.1474 (0.0160,0.3346)	0.1590 (0.0469,0.6325)	0.1827 (0.0669,0.2539)	0.720
C-4,6-dien-3	0.5965 (0.0523,1.3723)	0.4140 (0.0437,1.6658)	0.0794 (0.0376,0.6920)	0.418
6α-OHC	2.0950 (0.8482,4.3725)	1.5500 (0.8110,3.8313)	1.2473 (0.7175,2.2916)	0.422
27-CA	0.0033 (0.0026,0.0395)	0.0062 (0.0019,0.0138)	0.0038 (0.0023,0.0095)	0.939
7-HOCA	0.0004 (0.0002,0.0017)	0.0007 (0.0003,0.0018)	0.0006 (0.0003,0.0012)	0.550

Notes: MCI, mild cognitive impairment; MPS, MCI with possible sarcopenia; 4β-OHC: 4β-hydroxycholesterol; 4α-OHC: 4α-hydroxycholesterol; 7α/β-OHC: 7α/7β-hydroxycholesterol; 27-OHC: 27-hydroxycholesterol; 24S-OHC: 24S-hydroxycholesterol; 25-OHC: 25-hydroxycholesterol; 7K-25-OHC: 7-Keto,25-hydroxycholesterol; 7K-27-OHC: 7-Keto,27-hydroxycholesterol; 5α-6KC: 5α-hydroxy-6-keto cholesterol; 7α,25-diOHC: 7α,25-dihydroxycholesterol; 7-KC: 7-ketocholesterol; 24S,25-epoxy: 24S,25-epoxycholesterol; 5α,6α-epoxy: 5α,6α-epoxycholesterol; 5β,6β-epoxy: 5β,6β-epoxycholesterol; C4: 7α-hydroxy-4-cholesten-3-one; 15-HC: 5α-Cholest-8(14)-ene-3β,15α-diol; 3β,5α,6β-triol: Cholestαne-3β,5α,6β-triol; 3β,5α,6α-triol: Cholestαne-3β,5α,6α-triol; C-4,6-dien-3: Cholesta-4,6-dien-3-one; 6α-OHC: 6α-hydroxy-5α-cholestanol; 27-CA: 3β-hydroxy-5-cholestenoic acid; 7-HOCA: 7α-hydroxy-3-oxo-4-cholestenoic acid. * *p* < 0.05.

## Data Availability

The datasets used and/or analyzed during the current study are available from the corresponding author on reasonable request. The data are not available publicly due to privacy.

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
