# Peer review of "The Link Between Oxysterols and Gut Microbiota in the Co-Dysfunction of Cognition and Muscle"

_nutrients, 2025, doi:10.3390/nu17071277_

Round 1
Reviewer 1 Report
Comments and Suggestions for Authors
The manuscript presents results of an interesting study showing an association between serum oxysterols and gut microbiota in mild cognitive impairment individuals with or without possible sarcopenia. The study was performed on a representative group of 1035 individuals, among them 435 with mild cognitive impairment and 336 persons with mild cognitive impairment with possible sarcopenia. Detailed flow chart of detailed recruitment is presented. The presentation is proper, the conclusion is scientifically sound.
Remarks:
In the patients sarcopenia was manifested as decreased muscle strength without detectable changes in the muscle mass of the limbs. Is it possible that the function loss is due rather to disturbed information flow from the nervous system to the muscles than to muscle deterioration?
Can gut microbiota affect the oxysterol formation?
Please indicate which population was used for the estimation of correlations between the indexes of cognition or oxysterols and muscle: all persons studied or only MCI or MPS groups? Please indicate n.
The data for both genders were pooled. Were there any differences in the parameters and relationships between both genders and was there any age dependence?
References: Please report article titles in small letters in the middle.
Author Response
Comments 1:
In the patients sarcopenia was manifested as decreased muscle strength without detectable changes in the muscle mass of the limbs. Is it possible that the function loss is due rather to disturbed information flow from the nervous system to the muscles than to muscle deterioration?
Response 1:
Thank you for raising this insightful perspective.
The neural-muscle communication system is mediated by the neuromuscular junction (NMJ), which serves as the critical interface. As a highly specialized chemical synapse, the NMJ ensures faithful conversion of minute presynaptic motor neuron action potentials into robust postsynaptic muscle fiber contractions. Mechanistically, NMJ transmission deficits directly compromise force generation capacity [1].
During aging, NMJ undergoes structural remodeling to maintain neural-muscle connectivity. Agrin, a heparan sulfate proteoglycan, plays a pivotal role in organizing and stabilizing postsynaptic NMJ architecture. This age-related remodeling process involves neurotrypsin-mediated proteolytic cleavage of agrin, generating 22-kDa C-terminal agrin fragments (CAF) that exhibit measurable serum presence. Agrin fragmentation inactivates cholinergic receptors within presynaptic membranes, initiating a cascade of neuromuscular denervation and subsequent muscle fiber atrophy. This pathophysiological progression manifests as progressive loss of muscle mass accompanied by functional decline - characteristic hallmarks of sarcopenia. The quantifiable nature of circulating CAF positions it as a promising early diagnostic biomarker for sarcopenia [2].
On this basis, we added the corresponding contents to the discussion:
“In addition to the aforementioned assessment of serum biomarkers and muscle strength, muscle mass was also quantified. However, no intergroup differences were observed in appendicular muscle mass, suggesting that muscle functional decline may stem from impaired neural-to-muscle communication rather than muscle mass deterioration. The neural-muscular communication system is mediated by the neuromuscu-lar junction (NMJ), a highly specialized chemical synapse that converts presynaptic motor neuron action potentials into postsynaptic muscle fiber contractions [36]. Age-related NMJ remodeling involves proteolytic cleavage of Agrin into C-terminal Agrin fragments (CAF), which inactivates cholinergic receptors in presynaptic membranes and promotes denervated muscle fiber accumulation – a hallmark feature of sarcopenia. This mechanistic pathway supports serum CAF as a potential early diagnostic biomarker for sarcopenia [37]. Meta-analytical evidence demonstrates significant correlations between CAF levels and sarcopenia-related parameters including appendicular lean mass, handgrip strength, and gait velocity [38]. The present study revealed significantly elevated CAF levels in both MCI and MPS groups, indicating substantial NMJ dysfunction in populations with cognitive impairment and muscular deterioration, which may underlie observed alterations in muscle performance. These findings underscore the importance of targeting NMJ structural integrity and func-tional capacity as critical therapeutic strategies for sarcopenia prevention and man-agement.” [Page 20, Line 494~511]
Reference:
[1/36]. Arnold WD, Clark BC. Neuromuscular junction transmission failure in aging and sarcopenia: The nexus of the neurological and muscular systems. Ageing Res Rev. 2023 Aug;89:101966. doi: 10.1016/j.arr.2023.101966. Epub 2023 Jun 1. PMID: 37270145; PMCID: PMC10847753.
[2/37]. Fatima R, Kim Y, Baek S, Suram RP, An SL, Hong Y. C-Terminal Agrin Fragment as a Biomarker for Sarcopenia: A Systematic Review and Meta-Analysis. J Cachexia Sarcopenia Muscle. 2025 Feb;16(1):e13707. doi: 10.1002/jcsm.13707. PMID: 39887577; PMCID: PMC11780277.
[38]. Monti E, Sarto F, Sartori R, Zanchettin G, Löfler S, Kern H, Narici MV, Zampieri S. C-terminal agrin fragment as a biomarker of muscle wasting and weakness: a narrative review. J Cachexia Sarcopenia Muscle. 2023 Apr;14(2):730-744. doi: 10.1002/jcsm.13189. Epub 2023 Feb 11. PMID: 36772862; PMCID: PMC10067498.
Comments 2:
Can gut microbiota affect the oxysterol formation?
Response 2:
Regarding the relationship between gut microbiota and oxysterols, our research group has previously conducted multiple studies to explore this interaction.
On one hand, oxysterols, primarily 27-hydroxycholesterol (27-OHC), can influence the abundance and composition of gut microbiota [1-3]. On the other hand, gut microbiota also modulates oxysterol levels. In our prior work, intervention with Roseburia (a gut microbial genus) was shown to reverse 27-OHC-induced increases in cerebral 27-OHC and decreases in 24S-hydroxycholesterol (24S-OHC) [4].
Beyond our team’s findings, Sun et al. demonstrated that disrupting gut microbiota with antibiotics in hamsters upregulated CYP7B1. While their study focused on the role of CYP7B1 in the alternative bile acid synthesis pathway, it is noteworthy that CYP7B1 is also involved in oxysterol metabolism (e.g., converting 27-OHC to 27-carboxycholic acid [27-CA] and 7-hydroxy-3-oxocholest-4-en-26-oic acid [7-HOCA]). This suggests that gut microbiota dysbiosis may alter oxysterol production and metabolism by regulating the expression of oxysterol-metabolizing enzymes [5].
In summary, gut microbiota and oxysterols reciprocally influence each other, synergistically contributing to disease progression.
References:
[1]. Wang Y, An Y, Ma W, Yu H, Lu Y, Zhang X, Wang Y, Liu W, Wang T, Xiao R. 27-Hydroxycholesterol contributes to cognitive deficits in APP/PS1 transgenic mice through microbiota dysbiosis and intestinal barrier dysfunction. J Neuroinflammation. 2020 Jun 27;17(1):199. doi: 10.1186/s12974-020-01873-7. PMID: 32593306; PMCID: PMC7321549.
[2]. Hao L, Wang L, Ju M, Feng W, Guo Z, Sun X, Xiao R. 27-Hydroxycholesterol impairs learning and memory ability via decreasing brain glucose uptake mediated by the gut microbiota. Biomed Pharmacother. 2023 Dec;168:115649. doi: 10.1016/j.biopha.2023.115649. Epub 2023 Oct 6. PMID: 37806088.
[3]. Wang T, Hao L, Yang K, Feng W, Guo Z, Liu M, Xiao R. Fecal microbiota transplantation derived from mild cognitive impairment individuals impairs cerebral glucose uptake and cognitive function in wild-type mice: Bacteroidetes and TXNIP-GLUT signaling pathway. Gut Microbes. 2024 Jan-Dec;16(1):2395907. doi: 10.1080/19490976.2024.2395907. Epub 2024 Sep 12. PMID: 39262376; PMCID: PMC11404573.
[4]. Sun X, Zhou C, Ju M, Feng W, Guo Z, Qi C, Yang K, Xiao R. Roseburia intestinalis Supplementation Could Reverse the Learning and Memory Impairment and m6A Methylation Modification Decrease Caused by 27-Hydroxycholesterol in Mice. Nutrients. 2024 Apr 26;16(9):1288. doi: 10.3390/nu16091288. PMID: 38732535; PMCID: PMC11085097.
[5]. Sun L, Pang Y, Wang X, Wu Q, Liu H, Liu B, Liu G, Ye M, Kong W, Jiang C. Ablation of gut microbiota alleviates obesity-induced hepatic steatosis and glucose intolerance by modulating bile acid metabolism in hamsters. Acta Pharm Sin B. 2019 Jul;9(4):702-710. doi: 10.1016/j.apsb.2019.02.004. Epub 2019 Feb 16. PMID: 31384531; PMCID: PMC6664038.
Comments 3:
Please indicate which population was used for the estimation of correlations between the indexes of cognition or oxysterols and muscle: all persons studied or only MCI or MPS groups? Please indicate n.
Response 3:
Thank you for your professional reminder. The correlation analysis uses the data of all the subjects for analysis. At your suggestion, we have listed all the tested n values:
- Table 1:
General characteristics (n=1035)
Serum lipids (n=1032)
Dietary nutrients intake (n=957)
- Table 2:
Global cognition (n=1035)
Multidimensional cognition (n=1002)
- Table 3:
Muscle function (n=1023)
Muscle mass (n=528)
- To further verify the alterations of serum biomarkers of cognition and muscle, the concentrations of Nfl, Aβ1-42, APP, BDNF, irisin, and CAF were conducted by ELISA (n=15~20/group). [Page 10, Line 281~283]
- Table 4:
Control (n=29)
MCI (n=47)
MPS (n=27)
- To investigate the alternation of gut microbiota in the comorbidity, 16S rDNA sequencing was conducted (n=20/ group). [Page 13, Line 310~311]
- Spearman’s rank coefficients were calculated to estimate the correlation between cognition and muscle indicators based on the related data of all the three groups. [Page 15, Line 351~352]
Comments 4:
The data for both genders were pooled. Were there any differences in the parameters and relationships between both genders and was there any age dependence?
Response 4:
Thank you for your professional advice. We have made a hierarchical analysis of the cognitive and muscle testing functions by sex and age, and revised Table 2 and Table 3 as follows:
“A gender-stratified analysis was conducted to investigate differential cognitive performance patterns. In female participants, the control, MCI, and MPS groups ex-hibited statistically significant stepwise declines across multiple cognitive assessments (MMSE, MoCA, AVLT-SR, AVLT-LR). This graded pattern was not observed in males, where only reduced performance in both MCI and MPS groups relative to controls reached significance. The differences in DSF and TMTB performance observed in both males and females mirrored the patterns identified in the general population analysis. Notably, male participants demonstrated additional distinct characteristics: the MPS group showed significantly lower DSB and SDMT scores compared to both MCI and control groups, while displaying elevated TMTA scores. In contrast, female partici-pants exhibited progressive cognitive decline patterns, with both MCI and MPS groups demonstrating reduced DSB performance compared to controls, and SDMT scores showing a significant stepwise decrease across the three groups.
Age-stratified analyses demonstrated distinct cognitive patterns while maintain-ing consistency with overall trends. Both age subgroups (<65 and ≥65 years) exhibited significant progressive declines in SDMT scores accompanied by progressive elevation in TMTB performance, mirroring population-level observations. For MoCA and AVLT-SR assessments, while MCI and MPS groups showed reduced performance compared to controls across age strata, no significant differences emerged between MCI and MPS groups themselves. Notably, age-specific divergences were observed: In the <65y subgroup, DSB displayed significant stepwise deterioration, with concurrent reductions in MMSE, AVLT-LR, and DSF scores in both MCI and MPS groups relative to controls, while TMTA scores were specifically elevated in the MPS group. Conversely, the ≥65y subgroup revealed progressive improvement in MMSE and AVLT-LR scores across diagnostic groups, contrasted by significant stepwise worsening in TMTA per-formance. Both MCI and MPS groups maintained reduced DSB compared to controls in older adults, whereas DSF deficits became exclusive to the MPS group in this age stratum.” [Page 6~7, Line 232~258]
Table 2. Cognitive test scores.
|
Index |
Control |
MCI |
MPS |
P |
|
Global cognition (n=1035) |
||||
|
MMSE |
29 ± 2 |
27 ± 3 |
27 ± 3 |
<0.001* |
|
Male |
29 ± 2 |
28 ± 2 |
27 ± 2 |
<0.001* |
|
Famale |
29 ± 3 |
28 ± 2 |
28 ± 2 |
<0.001* |
|
<65y |
29 ± 1 |
28 ± 2 |
28 ± 2 |
<0.001* |
|
≥65y |
29 ± 2 |
29 ± 2 |
27 ± 1 |
<0.001* |
|
MoCA |
25 ± 2 |
22 ± 3 |
21 ± 4 |
<0.001* |
|
Male |
25 ± 2 |
23 ± 2.5 |
23 ± 3 |
<0.001* |
|
Famale |
25 ± 2 |
23 ± 3 |
22 ± 3 |
<0.001* |
|
<65y |
25 ± 2 |
23 ± 3 |
22 ± 3.5 |
<0.001* |
|
≥65y |
26 ± 2 |
23 ± 3 |
23 ± 2 |
<0.001* |
|
Multidimensional cognition (n=1002) |
||||
|
AVLT-SR |
6 ± 4 |
5 ± 4 |
5 ± 3 |
<0.001* |
|
Male |
5.5 ± 3 |
5 ± 3 |
4 ± 2 |
<0.001* |
|
Famale |
6 ± 3 |
5 ± 3.75 |
6 ± 3 |
<0.001* |
|
<65y |
6 ± 3 |
5 ± 4 |
5 ± 3 |
<0.001* |
|
≥65y |
6 ± 3 |
4 ± 5 |
4 ± 3 |
<0.001* |
|
AVLT-LR |
5 ± 4 |
4 ± 4 |
4 ± 3 |
<0.001* |
|
Male |
4 ± 3.75 |
4 ± 3 |
3 ± 3 |
<0.001* |
|
Famale |
6 ± 4 |
4 ± 3 |
4 ± 4 |
<0.001* |
|
<65y |
5 ± 4 |
4 ± 3 |
3.5 ± 4.75 |
<0.001* |
|
≥65y |
5.5 ± 3 |
3 ± 4 |
4 ± 3 |
<0.001* |
|
SDMT |
36 ± 17 |
35 ± 13.5 |
31 ± 12 |
<0.001* |
|
Male |
36 ± 13 |
35 ± 13 |
30.5 ± 10 |
<0.001* |
|
Famale |
38 ± 20 |
35 ± 14.75 |
32 ± 16 |
<0.001* |
|
<65y |
37 ± 15 |
36 ± 14.25 |
31 ± 14 |
<0.001* |
|
≥65y |
35 ± 19.5 |
31.5 ± 8.75 |
31 ± 12 |
<0.001* |
|
DSF |
8 ± 2 |
7 ± 1 |
8 ± 1 |
<0.001* |
|
Male |
8 ± 2 |
8 ± 1 |
8 ± 1 |
0.001 |
|
Famale |
8 ± 2 |
7 ± 2 |
8 ± 2 |
<0.001* |
|
<65y |
8 ± 2 |
7 ± 1 |
8 ± 2 |
<0.001* |
|
≥65y |
8 ± 1 |
7 ± 1.25 |
8 ± 1 |
0.015 |
|
DSB |
4 ± 1 |
4 ± 1 |
4 ± 1 |
<0.001* |
|
Male |
4 ± 2 |
4 ± 2 |
4 ± 1 |
0.004 |
|
Famale |
4 ± 1 |
4 ± 1 |
4 ± 1 |
<0.001* |
|
<65y |
4 ± 1 |
4 ± 1.25 |
4 ± 1 |
<0.001* |
|
≥65y |
4 ± 2 |
3 ± 1 |
4 ± 1 |
<0.001* |
|
TMTA |
60 ± 30 |
62 ± 35 |
70 ± 39 |
<0.001* |
|
Male |
57 ± 18.25 |
59 ± 26.5 |
67 ± 28 |
<0.001* |
|
Famale |
62 ± 34 |
64.5 ± 38 |
76 ± 39 |
<0.001* |
|
<65y |
60 ± 29 |
62 ± 34.75 |
70.5 ± 39 |
<0.001* |
|
≥65y |
57.5 ± 37.5 |
66.5 ± 25.75 |
68 ± 35 |
<0.001* |
|
TMTB |
144 ± 72 |
161 ± 90 |
186 ± 94 |
<0.001* |
|
Male |
142 ± 60.25 |
156 ± 88.5 |
175.5 ± 90.75 |
<0.001* |
|
Famale |
147 ± 75 |
170 ± 99.75 |
190 ± 103 |
<0.001* |
|
<65y |
140 ± 72 |
160 ± 90.25 |
190.5 ± 93 |
<0.001* |
|
≥65y |
149 ± 75.75 |
170.5 ± 108.25 |
160 ± 83 |
<0.001* |
|
SCWT-TIE |
36 ± 13 |
35 ± 18.5 |
36 ± 18 |
0.774 |
|
Male |
40 ± 18 |
42 ± 23.5 |
43.5 ± 21.5 |
0.370 |
|
Famale |
34 ± 13 |
33 ± 15 |
32 ± 16 |
0.539 |
|
<65y |
35 ± 13 |
34 ± 17 |
35 ± 18 |
0.807 |
|
≥65y |
39.5 ± 18 |
39.5 ± 19 |
42 ± 22 |
0.723 |
|
SCWT-RIE |
0 ± 2 |
0 ± 2 |
0 ± 2 |
0.848 |
|
Male |
0 ± 2 |
0 ± 2 |
0 ± 3.75 |
0.508 |
|
Famale |
0 ± 2 |
0 ± 3 |
0 ± 2 |
0.351 |
|
<65y |
0 ± 2 |
0 ± 2 |
0 ± 2.75 |
0.932 |
|
≥65y |
0 ± 1.75 |
1 ± 4 |
0 ± 1 |
0.158 |
|
Notes: MCI, mild cognitive impairment; MPS, MCI with possible sarcopenia; MMSE, mini-mental state examination; MoCA, Montreal cognitive assessment; AVLT-SR, auditory verbal learning test-short recall; AVLT-LR, auditory verbal learning test-long recall; SDMT, symbol digit modalities test; DSF, digit span forwards, DSB, digit span backwards; TMT-A, trail making test-A; TMT-B, trail making test-B; SCWT-TIE, Stroop color word test-time interfered effects; SCWT-RIE, Stroop color word test-reaction interfered effects. *P<0.05. |
||||
“Stratified analyses were extended to physical performance measures, revealing sex- and age-specific patterns. Both 5-time chair stand test and handgrip strength demonstrated subgroup differences across gender and age strata that mirrored those observed in the general population. Notably, sex-specific disparities in gait speed between control and MCI groups emerged exclusively in male participants. Furthermore, age-stratified analyses identified distinctive patterns in older adults: within the ≥65y subgroup, g6-metre walk in the MCI group was significantly reduced compared to both control and MPS groups, whereas such intergroup differences were not observed in younger participants.” [Page 8, Line 269~277]
Table 3. Muscle mass and function.
|
Index |
Control |
MCI |
MPS |
P |
|
Muscle function (n=1023) |
||||
|
5-time chair stand test (s) |
9.31 ± 2.43 |
9.70 ± 2.55 |
13.24 ± 2.77 |
<0.001* |
|
Male |
8.99 ± 2.27 |
9.61 ± 2.36 |
13.59 ± 2.84 |
<0.001* |
|
Famale |
9.40 ± 2.55 |
9.75 ± 2.66 |
12.98 ± 2.79 |
<0.001* |
|
<65y |
9.24 ± 2.32 |
9.59 ± 2.58 |
12.75 ± 2.80 |
<0.001* |
|
≥65y |
9.35 ± 2.44 |
9.80 ± 2.42 |
13.52 ± 2.99 |
<0.001* |
|
Handgrip strength (kg) |
26.10 ±13.00 |
25.60 ± 11.43 |
22.40 ± 10.35 |
<0.001* |
|
Male |
37.00 ± 8.60 |
35.35 ± 8.28 |
31.10 ± 7.15 |
<0.001* |
|
Famale |
22.95 ± 5.40 |
22.70 ± 4.65 |
18.90 ± 6.50 |
<0.001* |
|
<65y |
25.50 ± 9.27 |
25.40 ± 11.00 |
22.50 ± 9.70 |
<0.001* |
|
≥65y |
29.50 ± 13.40 |
26.90 ± 12.10 |
22.30 ± 11.40 |
<0.001* |
|
6-metre walk (m/s) |
1.13 ± 0.44 |
1.02 ± 0.45 |
1.01 ± 0.30 |
0.013* |
|
Male |
0.83 ± 0.21 |
0.82 ± 0.16 |
0.94 ± 0.20 |
0.031* |
|
Famale |
0.85 ± 0.23 |
0.84 ± 0.23 |
0.95 ± 0.25 |
0.306 |
|
<65y |
0.85 ± 0.24 |
0.83 ± 0.15 |
0.89 ± 0.19 |
0.056 |
|
≥65y |
0.83 ± 0.19 |
0.83 ± 0.22 |
0.97 ± 0.27 |
<0.001* |
|
Muscle mass (n=528) |
||||
|
Waistline (cm) |
85.00 ± 10.50 |
84.00 ± 11.13 |
86.00 ± 12.55 |
0.347 |
|
Male |
90.10 ± 7.81 |
88.08 ± 7.88 |
88.39 ± 7.65 |
0.776 |
|
Famale |
83.00 ± 10.13 |
82.00 ± 10.50 |
84.00 ± 11.00 |
0.256 |
|
<65y |
83.50 ± 13.55 |
83.00 ± 11.00 |
84.00 ± 13.00 |
0.631 |
|
≥65y |
85.50 ± 10.00 |
85.00 ± 11.00 |
87.00 ± 11.50 |
0.306 |
|
Hipline (cm) |
97.00 ± 8.75 |
96.00 ± 7.00 |
97.00 ± 8.00 |
0.488 |
|
Male |
101.00 ± 6.60 |
98.00 ± 5.80 |
98.00 ± 8.00 |
0.330 |
|
Famale |
96.40 ± 7.78 |
95.00 ± 7.13 |
96.00 ± 8.00 |
0.569 |
|
<65y |
97.00 ± 9.00 |
96.00 ± 5.50 |
96.00 ± 9.00 |
0.644 |
|
≥65y |
97.50 ± 7.50 |
96.00 ± 8.00 |
97.00 ± 7.00 |
0.443 |
|
Calf circumference (cm) |
34.50 ± 4.10 |
34.00 ± 3.25 |
34.00 ± 3.50 |
0.425 |
|
Male |
36.00 ± 4.30 |
35.75 ± 4.88 |
35.50 ± 3.50 |
0.440 |
|
Famale |
33.50 ± 4.08 |
33.50 ± 3.00 |
33.00 ± 3.00 |
0.505 |
|
<65y |
34.25 ± 4.35 |
34.00 ± 3.00 |
34.00 ± 3.50 |
0.194 |
|
≥65y |
34.50 ± 3.80 |
34.00 ± 4.00 |
34.00 ± 4.00 |
0.663 |
|
Muscle mass of right upper limb (kg) |
2.14 ± 0.92 |
2.16 ± 0.79 |
2.18 ± 0.93 |
0.557 |
|
Male |
2.93 ± 0.71 |
2.90 ± 0.59 |
2.91 ± 0.56 |
0.839 |
|
Famale |
2.01 ± 0.35 |
2.00 ± 0.31 |
1.92 ± 0.46 |
0.079 |
|
<65y |
2.09 ± 0.64 |
2.13 ± 0.70 |
2.04 ± 0.76 |
0.150 |
|
≥65y |
2.38 ± 0.96 |
2.31 ± 0.86 |
2.25 ± 0.93 |
0.817 |
|
Muscle mass of left upper limb (kg) |
2.12 ± 0.89 |
2.12 ± 0.76 |
2.12 ± 0.90 |
0.658 |
|
Male |
2.93 ± 0.43 |
2.87 ± 0.40 |
2.84 ± 0.46 |
0.851 |
|
Famale |
1.97 ± 0.38 |
1.97 ± 0.28 |
1.88 ± 0.44 |
0.120 |
|
<65y |
2.06 ± 0.65 |
2.10 ± 0.67 |
1.99 ± 0.70 |
0.146 |
|
≥65y |
2.35 ± 0.84 |
2.23 ± 0.79 |
2.23 ± 0.98 |
0.830 |
|
Muscle mass of right lower limb (kg) |
6.37 ± 2.15 |
6.44 ± 1.96 |
6.37 ± 2.35 |
0.491 |
|
Male |
8.39 ± 1.37 |
8.07 ± 1.24 |
8.10 ± 1.32 |
0.354 |
|
Famale |
6.05 ± 0.83 |
5.97 ± 0.96 |
5.67 ± 1.03 |
0.066 |
|
<65y |
6.22 ± 1.35 |
6.34 ± 1.67 |
6.05 ± 1.98 |
0.235 |
|
≥65y |
7.05 ± 2.66 |
6.66 ± 2.08 |
6.55 ± 2.66 |
0.818 |
|
Muscle mass of left lower limb (kg) |
6.29 ± 2.10 |
6.43 ± 2.02 |
6.38 ± 2.24 |
0.465 |
|
Male |
8.25 ± 1.55 |
8.04 ± 1.24 |
8.03 ± 1.32 |
0.412 |
|
Famale |
5.99 ± 0.78 |
5.93 ± 0.95 |
5.70 ± 1.04 |
0.061 |
|
<65y |
6.22 ± 1.48 |
6.32 ± 1.67 |
6.06 ± 1.75 |
0.210 |
|
≥65y |
7.11 ± 2.38 |
6.70 ± 2.20 |
6.57 ± 2.50 |
0.830 |
|
limbs ASM |
17.08 ± 5.78 |
17.10 ± 5.51 |
17.11 ± 6.58 |
0.512 |
|
Male |
22.49 ± 3.71 |
21.87 ± 3.77 |
21.95 ± 3.49 |
0.542 |
|
Famale |
15.74 ± 2.44 |
16.01 ± 2.49 |
15.07 ± 2.89 |
0.054 |
|
<65y |
16.66 ± 3.85 |
16.86 ± 4.41 |
16.08 ± 5.48 |
0.199 |
|
≥65y |
18.77 ± 6.75 |
18.03 ± 6.05 |
17.43 ± 7.11 |
0.848 |
|
ASM correction |
6.62 ± 1.44 |
6.64 ± 1.22 |
6.63 ± 1.48 |
0.679 |
|
Male |
7.65 ± 0.83 |
7.51 ± 0.89 |
7.69 ± 0.96 |
0.900 |
|
Famale |
6.23 ± 0.80 |
6.31 ± 0.66 |
6.11 ± 0.76 |
0.081 |
|
<65y |
6.40 ± 1.14 |
6.51 ± 1.19 |
6.40 ± 1.18 |
0.190 |
|
≥65y |
6.98 ± 1.43 |
6.75 ± 1.30 |
6.70 ± 1.57 |
0.758 |
|
Upper arm circumference (cm) |
30.60 ± 4.00 |
30.40 ± 3.50 |
30.40 ± 3.63 |
0.801 |
|
Male |
31.50 ± 2.30 |
31.45 ± 3.52 |
31.60 ± 3.60 |
0.940 |
|
Famale |
29.90 ± 3.60 |
29.60 ± 2.80 |
29.90 ± 3.50 |
0.936 |
|
<65y |
30.25 ± 4.55 |
30.50 ± 3.00 |
29.90 ± 3.75 |
0.673 |
|
≥65y |
31.00 ± 2.90 |
30.40 ± 3.70 |
30.90 ± 3.60 |
0.348 |
|
Fat-free circumference of upper arm (cm) |
24.10 ± 3.00 |
23.90 ± 2.73 |
24.10 ± 3.25 |
0.954 |
|
Male |
26.30 ± 1.20 |
26.00 ± 2.50 |
26.20 ± 2.45 |
0.966 |
|
Famale |
23.20 ± 1.85 |
23.30 ± 1.60 |
23.30 ± 2.10 |
0.831 |
|
<65y |
23.40 ± 2.90 |
23.90 ± 2.30 |
23.80 ± 2.85 |
0.306 |
|
≥65y |
24.60 ± 2.70 |
24.00 ± 3.00 |
24.30 ± 3.10 |
0.565 |
|
Notes: MCI, mild cognitive impairment; MPS, MCI with possible sarcopenia; ASM, appendicular lean mass. *P<0.05. |
||||
Comments 5:
References: Please report article titles in small letters in the middle.
Response 5:
We sincerely apologize for the inconsistency in the reference formatting. Thank you for highlighting this issue. We have now carefully revised all reference titles to ensure they strictly follow sentence case (only the first word and proper nouns capitalized, with all other words in lowercase).
Reviewer 2 Report
Comments and Suggestions for Authors
The present work stresses the levels of oxysterols measured in serum and gut microbiota characteristics in MCI individuals with or without possible sarcopenia. This is a very interesting approach combining anthropometric data along with biomarkers related to cognition and muscle functionality. Some comments could be taken into account for improvement.
The authors should provide details regarding the dietary, for example the Food Frequency Questionnaire was comprised by 33 food items, please explain.
Can the authors provide a detail protocol of serum oxysterols quantification performed by High‐Performance Liquid Chromatography tandem Mass Spectrometry, for example mobile phase, time of elution, internal standards, calibration curves; Also, spectra from HPLC and especially for MS detecting the levels of the oxysterols could be helpful and should be included.
Did the authors measure only Aβ 1-42 or Aβ40 (total amyloid beta); Please explain in detail the analytical procedure for ELISA quantification for Aβ and the other serum biomarkers.
The present work includes individuals with mild cognitive impairment and individuals in MCI with possible sarcopenia (MPS) and compared the levels of serum biomarkers with control group. The control group is healthy individuals or not? I am wondering as most of the biomarkers such as amyloid beta, APP or brain‐derived neurotrophic factor were not found statistically different between control and MCI or MPS group.
The authors should increase the quality of Figure 3, Figure 4 and Figure 5, most of the numbers and the provided information is not adequate.
Limitations of the present process and some future directions could be also discussed.
Author Response
Comments 1:
The authors should provide details regarding the dietary, for example the Food Frequency Questionnaire was comprised by 33 food items, please explain.
Response 1:
Thank you for your valuable feedback.
As requested, we have comprehensively supplemented the 33 dietary content items about dietary into Table S1 in the Supplementary Materials, and marked in the section of Materials and Methods in the manuscript, as follows:
The semiquantitative FFQ was comprised of 33 food items, of which the frequency (day, week, month, year) and amount (in grams or millilitre) were provided separately within 1 year before the survey (The details were shown in Table S1). [Page 3, Line 110~112]
|
Table S1 Composition of Food Frequency Questionnaire |
|||||
|
Food type |
frequency of eating |
Average weight per serving (50g) |
|||
|
Day |
Week |
Month |
Year |
||
|
1. rice |
|
|
|
|
|
|
2. wheat flour |
|
|
|
|
|
|
3. coarse cereals |
|
|
|
|
|
|
4. potatoes and yam |
|
|
|
|
|
|
5. fried pasta |
|
|
|
|
|
|
6. pork |
|
|
|
|
|
|
7. beef |
|
|
|
|
|
|
8. mutton |
|
|
|
|
|
|
9. chicken |
|
|
|
|
|
|
10. duck |
|
|
|
|
|
|
11. donkey meat |
|
|
|
|
|
|
12. visceral food |
|
|
|
|
|
|
13. other meat |
|
|
|
|
|
|
14. aquatic product |
|
|
|
|
|
|
15. milk |
|
|
|
|
|
|
16. milk powder |
|
|
|
|
|
|
17. cheese |
|
|
|
|
|
|
18. yogurt |
|
|
|
|
|
|
19. eggs |
|
|
|
|
|
|
20. tofu |
|
|
|
|
|
|
21. silk tofu |
|
|
|
|
|
|
22. tofu curd |
|
|
|
|
|
|
23. dried bean curd |
|
|
|
|
|
|
24. soya milk |
|
|
|
|
|
|
25. dried beans |
|
|
|
|
|
|
26. fresh vegetables |
|
|
|
|
|
|
27. dried vegetable |
|
|
|
|
|
|
28. salted vegetables |
|
|
|
|
|
|
29. pickles |
|
|
|
|
|
|
30. Chinese sauerkraut |
|
|
|
|
|
|
31. fresh fruit |
|
|
|
|
|
|
32. snack |
|
|
|
|
|
|
33. drinks |
|
|
|
|
|
|
Extra: cooking oil |
|
|
|
|
|
Comments 2:
Can the authors provide a detail protocol of serum oxysterols quantification performed by High‐Performance Liquid Chromatography tandem Mass Spectrometry, for example mobile phase, time of elution, internal standards, calibration curves; Also, spectra from HPLC and especially for MS detecting the levels of the oxysterols could be helpful and should be included.
Response 2:
Thank you for your professional advice. The detailed procedure of detecting serum oxysterols by UPLC-MS is as follows:
- The mobile phases: A and B consisted of (chloroform: methanol: ammonium hydroxide = 89.5: 10: 0.5) and (chloroform: methanol: ammonium hydroxide: water = 55: 39: 0.5: 5), respectively.
- The chromatographic elution gradient: starting at 5% B and maintained for 3 min, then increased to 40% B over 9 min, held at 40% B for 4 min, further increased to 70% B over 5 min, maintained at 70% B for 15 min, returned to 5% B over 3 min, and finally equilibrated for 6 min before the next injection. The injection volume was 5 μL, the column temperature was 25 °C, and the flow rate was 270 μL/min.
- Internal standards: D7-24-hydroxycholesterol, D7-7β-hydroxycholesterol, D6-25-hydroxycholesterol, D7-7-ketocholesterol, and D7-7α-hydroxy-4-cholesten-3-one.
- Mass spectrometry parameters: curtain gas, GAS1, and GAS2 at 20, 35, and 35 psi, respectively; ion source temperature at 400 °C; spray voltage at 5.5 kV; data acquired in electrospray ionization (ESI) mode; and oxysterols quantified using multiple reaction monitoring (MRM) mode.
The aforementioned analyses were performed by LipidALL Technologies Co., Ltd. Since the results provided by the company do not include calibration curves or spectral data, only the aforementioned UPLC-MS analytical information can be provided at this time.
We have also revised the manuscript as follows:
“The mobile phases A and B consisted of (chloroform: methanol: ammonium hydroxide = 89.5: 10: 0.5) and (chloroform: methanol: ammonium hydrox-ide: water = 55: 39: 0.5: 5), respectively. The chromatographic elution gradient was as follows: 5% B, 3 min → 5%~40% B, 9 min → 40% B, 4 min → 40%~ 70% B, 5 min → 70% B, 15 min → 70%~5% B, 3 min → equilibrated for 6 min before the next injection. A Phenomenex Luna Silica 3 µm column (internal diameter 150×2.0 mm; column tem-perature, 25 °C; flow rate, 270 μL/min) was used for the separation of oxysterols with the 5μL of sample volume. The Mass spectrometry parameters was: curtain gas, GAS1, and GAS2 at 20, 35, and 35 psi, respectively; ion source temperature at 400 °C; spray voltage at 5.5 kV. Eventually, the data was collected in electrospray ionization (ESI) mode, while the quantification of oxysterols was performed with the multiple reaction monitoring (MRM) mode.” [Page 5, Line 169~180]
Comments 3:
Did the authors measure only Aβ 1-42 or Aβ40 (total amyloid beta); Please explain in detail the analytical procedure for ELISA quantification for Aβ and the other serum biomarkers.
Response 3:
Thank you for raising these critical points. We have implemented the following revisions to address your concerns:
- In this study, Aβ1-42, which performed stronger neurotoxicity, was measured; therefore, all references to "Aβ" in the manuscript and Figure 2B have been explicitly revised to "Aβ1-42".
- The catalog number of the ELISA kit used for biomarker quantification has been added to the Materials and Methods section, as follows:
“Concentrations of Aβ1-42 (E-EL-H0543), amyloid precursor protein (APP, E-EL-H1216), neurofilament (Nfl, E-EL-H0741c), brain-derived neurotrophic factor (BDNF, E-EL-H0010), C-terminal agrin fragment (CAF, ELK9769), and irisin (E-EL-H5735) were evaluated by ELISA kit. Detailed experimental steps are shown in supplementary materials.” [Page 5, Line 181~185]
- A step-by-step detailed protocol, including sample preparation, incubation conditions, and data normalization procedures, is now available in Supplementary File S2.
Comments 4:
The present work includes individuals with mild cognitive impairment and individuals in MCI with possible sarcopenia (MPS) and compared the levels of serum biomarkers with control group. The control group is healthy individuals or not? I am wondering as most of the biomarkers such as amyloid beta, APP or brain‐derived neurotrophic factor were not found statistically different between control and MCI or MPS group.
Response 4:
As outlined in the flowchart, the control group was defined as healthy individuals with normal cognitive and muscular function based on cognitive assessments and muscular function evaluations. The absence of significant differences in several biomarkers between the control and case groups suggests that corresponding pathological alterations may not yet manifest in peripheral circulation. To address this limitation, the methodological improvements should be prioritized:
Given the high cost of ELISA assays, the current sample size for biomarker analysis (20~30 per group) was smaller than that for cognitive assessments. Future large-scale population studies with increased sample sizes are warranted to enhance statistical power and reliability.
This limitation has been supplemented in the discussion section:
“Firstly, owing to cost constraints, not all samples underwent oxysterol and biomarker analyses, which may limit the representativeness of these biochemical measures compared to the more comprehensively assessed cognitive and muscular outcomes. Future large-scale investigations are warranted to validate the reliability of these findings.” [Page 21, Line 560~564]
Comments 5:
The authors should increase the quality of Figure 3, Figure 4 and Figure 5, most of the numbers and the provided information is not adequate.
Response 5:
Thank you for your feedback on the quality and completeness of Figures 3-5. We apologize for the oversight and have revised these figures to address your concerns.
Specifically, the text and layout of Figures 3 and 4 have been rearranged. To improve clarity and resolution, Figure 5G and 5H have been separated into new Figures 6 and 7, respectively, reducing overcrowding and enhancing visual presentation. Text labels and annotations across all figures (3-5, 6-7) were repositioned to align with the manuscript template and improve readability. Additionally, figure captions and relevant text sections were updated to provide clearer context for the data.
Comments 6:
Limitations of the present process and some future directions could be also discussed.
Response 6:
Thank you for your constructive feedback. We have carefully incorporated your suggestions and have now added the study’s limitations and future research directions in the revised manuscript:
“This research has several limitations that should be acknowledged. Firstly, owing to cost constraints, not all samples underwent oxysterol and biomarker analyses, which may limit the representativeness of these biochemical measures compared to the more comprehensively assessed cognitive and muscular outcomes. Future large-scale investigations are warranted to validate the reliability of these findings. Secondly, the cohort was exclusively derived from a Chinese population, potentially restricting the generalizability of the observed cognitive-muscular dysfunction relationships to global populations with distinct genetic and environmental backgrounds. Multi-center co-horts involving diverse ethnicities are critical to elucidate the universality and mecha-nisms of cognitive-muscular comorbidities. Finally, as an observational study, our findings primarily establish associations between oxysterols, gut microbiota, and cog-nitive-muscular comorbidities. However, whether oxysterols mechanistically influence these comorbidities through gut microbial modulation remains speculative and re-quires validation via targeted mechanistic animal studies.” [Page 21~22, Line 560~573]
“In the future, priority should be given to eluci-dating causal pathways (e.g., gut-brain-muscle axis via animal models), integrating multi-omics (metabolomics / metagenomics) to identify biomarkers, and testing inter-ventions (probiotics / diet) to restore homeostasis. Expanding to diverse global cohorts will clarify universality of these findings, while translating discoveries into du-al-purpose diagnostic/therapeutic strategies could mitigate aging-related brain and muscle decline.” [Page 22, Line 578~583]
Reviewer 3 Report
Comments and Suggestions for Authors
The authors present an interesting study in which the relationship between oxysterols and the subsequent balance in gut microbiota with the clinical outcomes of cognitive impairment and sarcopenia is explored. Briefly, the authors recruited participants to be part of three distinct groups, control, mild cognitive impairment, and mild cognitive impairment with signs of sarcopenia, to undertake clinical evaluation and provide blood and stool samples. Ultimately, each group presented with distinct clinical differences, but also distinct and consistent changes in the levels of certain oxysterols and particular gut bacteria, essentially linking the levels of such with influencing cognitive and muscle capabilities. Ultimately, this article highlights the importance of these aspects in each category and identifies targets for possible therapeutic intervention and/or targeting.
In reviewing the article, I made a couple of observations. The following should be considered in any resubmission.
- Was there any protocol in place for the participants to abide by in advance of the stool sample being collected?
- How long were the samples in storage before they were analysed? Is it a case that each sample was read on collection, or were all the samples accumulated and then read at the same time? Do the authors believe the age of the sample may have impacted on the results obtained?
- It is assumed that in the various tables of data in which the data for each group is being presented, all participants are represented? Can the authors confirm whether any data sees a change in n-number from table to table? Similarly, it would be useful if the n-number was included on Figure 2 and Figure 5 where appropriate.
- The data in Figure 3 and Figure 4 and Figure 5 in particular are too small in scale, and the details are incredibly difficult to read. The authors should strongly consider revisiting the formatting of this figure and improving the scale of the writing in particular to ensure it is clearly legible.
- It would have been interesting to see analyses performed with respect to gender for example. Have the authors performed this out of curiosity?
Author Response
Comments 1:
Was there any protocol in place for the participants to abide by in advance of the stool sample being collected?
Response 1:
Thank you for emphasizing the importance of sample integrity. To ensure the reliability of our analyses, we implemented a standardized protocol for fecal sample preservation, as detailed in manuscript:
“To ensure the quality of fecal samples and the reliability of the data, a rigorous protocol was implemented: trained staff provided one-on-one guidance on sample collection using medical-standard containers, requiring a minimum broad bean-sized volume (~2 g) and storage at -80°C within 2 hours of collection. A single freezing protocol was implemented to avoid degradation caused by freeze-thaw cycles.” [Page 2, Line 86~90]
Comments 2:
How long were the samples in storage before they were analysed? Is it a case that each sample was read on collection, or were all the samples accumulated and then read at the same time? Do the authors believe the age of the sample may have impacted on the results obtained?
Response 2:
All samples were analyzed within 6 months post-collection. To ensure analytical consistency and minimize potential inter-batch variability, specimens were batched collectively and processed uniformly in a single assay run. This approach mitigates confounding effects from technical fluctuations across independent experimental batches, which could otherwise compromise the comparability of results.
During the analytical phase, samples were randomly selected for processing. As per your suggestion, we conducted a post hoc age analysis on the tested subset. The results revealed that the control group exhibited a higher mean age compared to the MCI group, whereas no significant age disparity was observed between the control and MPS groups. This suggests that age may confound the microbial differences between the control and MCI groups but does not affect comparisons involving the MPS group.
Notably, advanced age is an established risk factor for gut microbiota dysbiosis. Consequently, age-related microbial perturbations in older controls might attenuate inter-group differences (control vs. MCI), potentially masking true associations (i.e., false-negative risk). Importantly, however, the persistent microbial distinctions between control and MCI groups under this age bias imply that the observed differences likely reflect true pathological alterations rather than demographic artifacts.
Comments 3:
It is assumed that in the various tables of data in which the data for each group is being presented, all participants are represented? Can the authors confirm whether any data sees a change in n-number from table to table? Similarly, it would be useful if the n-number was included on Figure 2 and Figure 5 where appropriate.
Response 3:
Due to the limitation of testing cost, not everyone has done all the testing, so we marked all the tested N in the corresponding position in the text or table, as follows:
- Table 1:
General characteristics (n=1035)
Serum lipids (n=1032)
Dietary nutrients intake (n=957)
- Table 2:
Global cognition (n=1035)
Multidimensional cognition (n=1002)
- Table 3:
Muscle function (n=1023)
Muscle mass (n=528)
- To further verify the alterations of serum biomarkers of cognition and muscle, the concentrations of Nfl, Aβ1-42, APP, BDNF, irisin, and CAF were conducted by ELISA (n=15~20/group). [Page 10, Line 281~283]
- Table 4:
Control (n=29)
MCI (n=47)
MPS (n=27)
- To investigate the alternation of gut microbiota in the comorbidity, 16S rDNA sequencing was conducted (n=20/ group). [Page 13, Line 310~311]
- Spearman’s rank coefficients were calculated to estimate the correlation between cognition and muscle indicators based on the related data of all the three groups. [Page 15, Line 351~352]
Comments 4:
The data in Figure 3 and Figure 4 and Figure 5 in particular are too small in scale, and the details are incredibly difficult to read. The authors should strongly consider revisiting the formatting of this figure and improving the scale of the writing in particular to ensure it is clearly legible.
Response 4:
Thank you for highlighting the need to improve the formatting and readability of the figures. We have carefully revised the figures as suggested: Figures 3 and 4 have been re-adjusted in scale and layout to enhance clarity, and Figures 5G and 5H have been split into separate Figures 6 and 7 to eliminate overcrowding and ensure better visual resolution. All text labels and annotations were repositioned to align with the manuscript template and improve readability.
Comments 5:
It would have been interesting to see analyses performed with respect to gender for example. Have the authors performed this out of curiosity?
Response 5:
Thank you for your professional advice. We have made a hierarchical analysis of the cognitive and muscle testing functions by sex and age, and revised Table 2 and Table 3 as follows:
“A gender-stratified analysis was conducted to investigate differential cognitive performance patterns. In female participants, the control, MCI, and MPS groups ex-hibited statistically significant stepwise declines across multiple cognitive assessments (MMSE, MoCA, AVLT-SR, AVLT-LR). This graded pattern was not observed in males, where only reduced performance in both MCI and MPS groups relative to controls reached significance. The differences in DSF and TMTB performance observed in both males and females mirrored the patterns identified in the general population analysis. Notably, male participants demonstrated additional distinct characteristics: the MPS group showed significantly lower DSB and SDMT scores compared to both MCI and control groups, while displaying elevated TMTA scores. In contrast, female partici-pants exhibited progressive cognitive decline patterns, with both MCI and MPS groups demonstrating reduced DSB performance compared to controls, and SDMT scores showing a significant stepwise decrease across the three groups.
Age-stratified analyses demonstrated distinct cognitive patterns while maintain-ing consistency with overall trends. Both age subgroups (<65 and ≥65 years) exhibited significant progressive declines in SDMT scores accompanied by progressive elevation in TMTB performance, mirroring population-level observations. For MoCA and AVLT-SR assessments, while MCI and MPS groups showed reduced performance compared to controls across age strata, no significant differences emerged between MCI and MPS groups themselves. Notably, age-specific divergences were observed: In the <65y subgroup, DSB displayed significant stepwise deterioration, with concurrent reductions in MMSE, AVLT-LR, and DSF scores in both MCI and MPS groups relative to controls, while TMTA scores were specifically elevated in the MPS group. Conversely, the ≥65y subgroup revealed progressive improvement in MMSE and AVLT-LR scores across diagnostic groups, contrasted by significant stepwise worsening in TMTA per-formance. Both MCI and MPS groups maintained reduced DSB compared to controls in older adults, whereas DSF deficits became exclusive to the MPS group in this age stratum.” [Page 6~7, Line 232~258]
Table 2. Cognitive test scores.
|
Index |
Control |
MCI |
MPS |
P |
|
Global cognition (n=1035) |
||||
|
MMSE |
29 ± 2 |
27 ± 3 |
27 ± 3 |
<0.001* |
|
Male |
29 ± 2 |
28 ± 2 |
27 ± 2 |
<0.001* |
|
Famale |
29 ± 3 |
28 ± 2 |
28 ± 2 |
<0.001* |
|
<65y |
29 ± 1 |
28 ± 2 |
28 ± 2 |
<0.001* |
|
≥65y |
29 ± 2 |
29 ± 2 |
27 ± 1 |
<0.001* |
|
MoCA |
25 ± 2 |
22 ± 3 |
21 ± 4 |
<0.001* |
|
Male |
25 ± 2 |
23 ± 2.5 |
23 ± 3 |
<0.001* |
|
Famale |
25 ± 2 |
23 ± 3 |
22 ± 3 |
<0.001* |
|
<65y |
25 ± 2 |
23 ± 3 |
22 ± 3.5 |
<0.001* |
|
≥65y |
26 ± 2 |
23 ± 3 |
23 ± 2 |
<0.001* |
|
Multidimensional cognition (n=1002) |
||||
|
AVLT-SR |
6 ± 4 |
5 ± 4 |
5 ± 3 |
<0.001* |
|
Male |
5.5 ± 3 |
5 ± 3 |
4 ± 2 |
<0.001* |
|
Famale |
6 ± 3 |
5 ± 3.75 |
6 ± 3 |
<0.001* |
|
<65y |
6 ± 3 |
5 ± 4 |
5 ± 3 |
<0.001* |
|
≥65y |
6 ± 3 |
4 ± 5 |
4 ± 3 |
<0.001* |
|
AVLT-LR |
5 ± 4 |
4 ± 4 |
4 ± 3 |
<0.001* |
|
Male |
4 ± 3.75 |
4 ± 3 |
3 ± 3 |
<0.001* |
|
Famale |
6 ± 4 |
4 ± 3 |
4 ± 4 |
<0.001* |
|
<65y |
5 ± 4 |
4 ± 3 |
3.5 ± 4.75 |
<0.001* |
|
≥65y |
5.5 ± 3 |
3 ± 4 |
4 ± 3 |
<0.001* |
|
SDMT |
36 ± 17 |
35 ± 13.5 |
31 ± 12 |
<0.001* |
|
Male |
36 ± 13 |
35 ± 13 |
30.5 ± 10 |
<0.001* |
|
Famale |
38 ± 20 |
35 ± 14.75 |
32 ± 16 |
<0.001* |
|
<65y |
37 ± 15 |
36 ± 14.25 |
31 ± 14 |
<0.001* |
|
≥65y |
35 ± 19.5 |
31.5 ± 8.75 |
31 ± 12 |
<0.001* |
|
DSF |
8 ± 2 |
7 ± 1 |
8 ± 1 |
<0.001* |
|
Male |
8 ± 2 |
8 ± 1 |
8 ± 1 |
0.001 |
|
Famale |
8 ± 2 |
7 ± 2 |
8 ± 2 |
<0.001* |
|
<65y |
8 ± 2 |
7 ± 1 |
8 ± 2 |
<0.001* |
|
≥65y |
8 ± 1 |
7 ± 1.25 |
8 ± 1 |
0.015 |
|
DSB |
4 ± 1 |
4 ± 1 |
4 ± 1 |
<0.001* |
|
Male |
4 ± 2 |
4 ± 2 |
4 ± 1 |
0.004 |
|
Famale |
4 ± 1 |
4 ± 1 |
4 ± 1 |
<0.001* |
|
<65y |
4 ± 1 |
4 ± 1.25 |
4 ± 1 |
<0.001* |
|
≥65y |
4 ± 2 |
3 ± 1 |
4 ± 1 |
<0.001* |
|
TMTA |
60 ± 30 |
62 ± 35 |
70 ± 39 |
<0.001* |
|
Male |
57 ± 18.25 |
59 ± 26.5 |
67 ± 28 |
<0.001* |
|
Famale |
62 ± 34 |
64.5 ± 38 |
76 ± 39 |
<0.001* |
|
<65y |
60 ± 29 |
62 ± 34.75 |
70.5 ± 39 |
<0.001* |
|
≥65y |
57.5 ± 37.5 |
66.5 ± 25.75 |
68 ± 35 |
<0.001* |
|
TMTB |
144 ± 72 |
161 ± 90 |
186 ± 94 |
<0.001* |
|
Male |
142 ± 60.25 |
156 ± 88.5 |
175.5 ± 90.75 |
<0.001* |
|
Famale |
147 ± 75 |
170 ± 99.75 |
190 ± 103 |
<0.001* |
|
<65y |
140 ± 72 |
160 ± 90.25 |
190.5 ± 93 |
<0.001* |
|
≥65y |
149 ± 75.75 |
170.5 ± 108.25 |
160 ± 83 |
<0.001* |
|
SCWT-TIE |
36 ± 13 |
35 ± 18.5 |
36 ± 18 |
0.774 |
|
Male |
40 ± 18 |
42 ± 23.5 |
43.5 ± 21.5 |
0.370 |
|
Famale |
34 ± 13 |
33 ± 15 |
32 ± 16 |
0.539 |
|
<65y |
35 ± 13 |
34 ± 17 |
35 ± 18 |
0.807 |
|
≥65y |
39.5 ± 18 |
39.5 ± 19 |
42 ± 22 |
0.723 |
|
SCWT-RIE |
0 ± 2 |
0 ± 2 |
0 ± 2 |
0.848 |
|
Male |
0 ± 2 |
0 ± 2 |
0 ± 3.75 |
0.508 |
|
Famale |
0 ± 2 |
0 ± 3 |
0 ± 2 |
0.351 |
|
<65y |
0 ± 2 |
0 ± 2 |
0 ± 2.75 |
0.932 |
|
≥65y |
0 ± 1.75 |
1 ± 4 |
0 ± 1 |
0.158 |
|
Notes: MCI, mild cognitive impairment; MPS, MCI with possible sarcopenia; MMSE, mini-mental state examination; MoCA, Montreal cognitive assessment; AVLT-SR, auditory verbal learning test-short recall; AVLT-LR, auditory verbal learning test-long recall; SDMT, symbol digit modalities test; DSF, digit span forwards, DSB, digit span backwards; TMT-A, trail making test-A; TMT-B, trail making test-B; SCWT-TIE, Stroop color word test-time interfered effects; SCWT-RIE, Stroop color word test-reaction interfered effects. *P<0.05. |
||||
Stratified analyses were extended to physical performance measures, revealing sex- and age-specific patterns. Both 5-time chair stand test and handgrip strength demonstrated subgroup differences across gender and age strata that mirrored those observed in the general population. Notably, sex-specific disparities in gait speed between control and MCI groups emerged exclusively in male participants. Furthermore, age-stratified analyses identified distinctive patterns in older adults: within the ≥65y subgroup, g6-metre walk in the MCI group was significantly reduced compared to both control and MPS groups, whereas such intergroup differences were not observed in younger participants. [Page 8, Line 269~277]
Table 3. Muscle mass and function.
|
Index |
Control |
MCI |
MPS |
P |
|
Muscle function (n=1023) |
||||
|
5-time chair stand test (s) |
9.31 ± 2.43 |
9.70 ± 2.55 |
13.24 ± 2.77 |
<0.001* |
|
Male |
8.99 ± 2.27 |
9.61 ± 2.36 |
13.59 ± 2.84 |
<0.001* |
|
Famale |
9.40 ± 2.55 |
9.75 ± 2.66 |
12.98 ± 2.79 |
<0.001* |
|
<65y |
9.24 ± 2.32 |
9.59 ± 2.58 |
12.75 ± 2.80 |
<0.001* |
|
≥65y |
9.35 ± 2.44 |
9.80 ± 2.42 |
13.52 ± 2.99 |
<0.001* |
|
Handgrip strength (kg) |
26.10 ±13.00 |
25.60 ± 11.43 |
22.40 ± 10.35 |
<0.001* |
|
Male |
37.00 ± 8.60 |
35.35 ± 8.28 |
31.10 ± 7.15 |
<0.001* |
|
Famale |
22.95 ± 5.40 |
22.70 ± 4.65 |
18.90 ± 6.50 |
<0.001* |
|
<65y |
25.50 ± 9.27 |
25.40 ± 11.00 |
22.50 ± 9.70 |
<0.001* |
|
≥65y |
29.50 ± 13.40 |
26.90 ± 12.10 |
22.30 ± 11.40 |
<0.001* |
|
6-metre walk (m/s) |
1.13 ± 0.44 |
1.02 ± 0.45 |
1.01 ± 0.30 |
0.013* |
|
Male |
0.83 ± 0.21 |
0.82 ± 0.16 |
0.94 ± 0.20 |
0.031* |
|
Famale |
0.85 ± 0.23 |
0.84 ± 0.23 |
0.95 ± 0.25 |
0.306 |
|
<65y |
0.85 ± 0.24 |
0.83 ± 0.15 |
0.89 ± 0.19 |
0.056 |
|
≥65y |
0.83 ± 0.19 |
0.83 ± 0.22 |
0.97 ± 0.27 |
<0.001* |
|
Muscle mass (n=528) |
||||
|
Waistline (cm) |
85.00 ± 10.50 |
84.00 ± 11.13 |
86.00 ± 12.55 |
0.347 |
|
Male |
90.10 ± 7.81 |
88.08 ± 7.88 |
88.39 ± 7.65 |
0.776 |
|
Famale |
83.00 ± 10.13 |
82.00 ± 10.50 |
84.00 ± 11.00 |
0.256 |
|
<65y |
83.50 ± 13.55 |
83.00 ± 11.00 |
84.00 ± 13.00 |
0.631 |
|
≥65y |
85.50 ± 10.00 |
85.00 ± 11.00 |
87.00 ± 11.50 |
0.306 |
|
Hipline (cm) |
97.00 ± 8.75 |
96.00 ± 7.00 |
97.00 ± 8.00 |
0.488 |
|
Male |
101.00 ± 6.60 |
98.00 ± 5.80 |
98.00 ± 8.00 |
0.330 |
|
Famale |
96.40 ± 7.78 |
95.00 ± 7.13 |
96.00 ± 8.00 |
0.569 |
|
<65y |
97.00 ± 9.00 |
96.00 ± 5.50 |
96.00 ± 9.00 |
0.644 |
|
≥65y |
97.50 ± 7.50 |
96.00 ± 8.00 |
97.00 ± 7.00 |
0.443 |
|
Calf circumference (cm) |
34.50 ± 4.10 |
34.00 ± 3.25 |
34.00 ± 3.50 |
0.425 |
|
Male |
36.00 ± 4.30 |
35.75 ± 4.88 |
35.50 ± 3.50 |
0.440 |
|
Famale |
33.50 ± 4.08 |
33.50 ± 3.00 |
33.00 ± 3.00 |
0.505 |
|
<65y |
34.25 ± 4.35 |
34.00 ± 3.00 |
34.00 ± 3.50 |
0.194 |
|
≥65y |
34.50 ± 3.80 |
34.00 ± 4.00 |
34.00 ± 4.00 |
0.663 |
|
Muscle mass of right upper limb (kg) |
2.14 ± 0.92 |
2.16 ± 0.79 |
2.18 ± 0.93 |
0.557 |
|
Male |
2.93 ± 0.71 |
2.90 ± 0.59 |
2.91 ± 0.56 |
0.839 |
|
Famale |
2.01 ± 0.35 |
2.00 ± 0.31 |
1.92 ± 0.46 |
0.079 |
|
<65y |
2.09 ± 0.64 |
2.13 ± 0.70 |
2.04 ± 0.76 |
0.150 |
|
≥65y |
2.38 ± 0.96 |
2.31 ± 0.86 |
2.25 ± 0.93 |
0.817 |
|
Muscle mass of left upper limb (kg) |
2.12 ± 0.89 |
2.12 ± 0.76 |
2.12 ± 0.90 |
0.658 |
|
Male |
2.93 ± 0.43 |
2.87 ± 0.40 |
2.84 ± 0.46 |
0.851 |
|
Famale |
1.97 ± 0.38 |
1.97 ± 0.28 |
1.88 ± 0.44 |
0.120 |
|
<65y |
2.06 ± 0.65 |
2.10 ± 0.67 |
1.99 ± 0.70 |
0.146 |
|
≥65y |
2.35 ± 0.84 |
2.23 ± 0.79 |
2.23 ± 0.98 |
0.830 |
|
Muscle mass of right lower limb (kg) |
6.37 ± 2.15 |
6.44 ± 1.96 |
6.37 ± 2.35 |
0.491 |
|
Male |
8.39 ± 1.37 |
8.07 ± 1.24 |
8.10 ± 1.32 |
0.354 |
|
Famale |
6.05 ± 0.83 |
5.97 ± 0.96 |
5.67 ± 1.03 |
0.066 |
|
<65y |
6.22 ± 1.35 |
6.34 ± 1.67 |
6.05 ± 1.98 |
0.235 |
|
≥65y |
7.05 ± 2.66 |
6.66 ± 2.08 |
6.55 ± 2.66 |
0.818 |
|
Muscle mass of left lower limb (kg) |
6.29 ± 2.10 |
6.43 ± 2.02 |
6.38 ± 2.24 |
0.465 |
|
Male |
8.25 ± 1.55 |
8.04 ± 1.24 |
8.03 ± 1.32 |
0.412 |
|
Famale |
5.99 ± 0.78 |
5.93 ± 0.95 |
5.70 ± 1.04 |
0.061 |
|
<65y |
6.22 ± 1.48 |
6.32 ± 1.67 |
6.06 ± 1.75 |
0.210 |
|
≥65y |
7.11 ± 2.38 |
6.70 ± 2.20 |
6.57 ± 2.50 |
0.830 |
|
limbs ASM |
17.08 ± 5.78 |
17.10 ± 5.51 |
17.11 ± 6.58 |
0.512 |
|
Male |
22.49 ± 3.71 |
21.87 ± 3.77 |
21.95 ± 3.49 |
0.542 |
|
Famale |
15.74 ± 2.44 |
16.01 ± 2.49 |
15.07 ± 2.89 |
0.054 |
|
<65y |
16.66 ± 3.85 |
16.86 ± 4.41 |
16.08 ± 5.48 |
0.199 |
|
≥65y |
18.77 ± 6.75 |
18.03 ± 6.05 |
17.43 ± 7.11 |
0.848 |
|
ASM correction |
6.62 ± 1.44 |
6.64 ± 1.22 |
6.63 ± 1.48 |
0.679 |
|
Male |
7.65 ± 0.83 |
7.51 ± 0.89 |
7.69 ± 0.96 |
0.900 |
|
Famale |
6.23 ± 0.80 |
6.31 ± 0.66 |
6.11 ± 0.76 |
0.081 |
|
<65y |
6.40 ± 1.14 |
6.51 ± 1.19 |
6.40 ± 1.18 |
0.190 |
|
≥65y |
6.98 ± 1.43 |
6.75 ± 1.30 |
6.70 ± 1.57 |
0.758 |
|
Upper arm circumference (cm) |
30.60 ± 4.00 |
30.40 ± 3.50 |
30.40 ± 3.63 |
0.801 |
|
Male |
31.50 ± 2.30 |
31.45 ± 3.52 |
31.60 ± 3.60 |
0.940 |
|
Famale |
29.90 ± 3.60 |
29.60 ± 2.80 |
29.90 ± 3.50 |
0.936 |
|
<65y |
30.25 ± 4.55 |
30.50 ± 3.00 |
29.90 ± 3.75 |
0.673 |
|
≥65y |
31.00 ± 2.90 |
30.40 ± 3.70 |
30.90 ± 3.60 |
0.348 |
|
Fat-free circumference of upper arm (cm) |
24.10 ± 3.00 |
23.90 ± 2.73 |
24.10 ± 3.25 |
0.954 |
|
Male |
26.30 ± 1.20 |
26.00 ± 2.50 |
26.20 ± 2.45 |
0.966 |
|
Famale |
23.20 ± 1.85 |
23.30 ± 1.60 |
23.30 ± 2.10 |
0.831 |
|
<65y |
23.40 ± 2.90 |
23.90 ± 2.30 |
23.80 ± 2.85 |
0.306 |
|
≥65y |
24.60 ± 2.70 |
24.00 ± 3.00 |
24.30 ± 3.10 |
0.565 |
|
Notes: MCI, mild cognitive impairment; MPS, MCI with possible sarcopenia; ASM, appendicular lean mass. *P<0.05. |
||||
Round 2
Reviewer 3 Report
Comments and Suggestions for Authors
The authors have responded positively to my comments and the manuscript is much improved. I would like a small bit more detail on the collection of the stool sample however. When I requested the protocol, I was interested in whether individuals were subjected to any kind of fasting, or changes in dietary patterns, reduction in alcohol, medications, etc. , or whether any protocol at all was in place for these aspects and instead the samples represented a snapshot of the gut microbiota at random
Author Response
Comment 1: The authors have responded positively to my comments and the manuscript is much improved. I would like a small bit more detail on the collection of the stool sample however. When I requested the protocol, I was interested in whether individuals were subjected to any kind of fasting, or changes in dietary patterns, reduction in alcohol, medications, etc. , or whether any protocol at all was in place for these aspects and instead the samples represented a snapshot of the gut microbiota at random
Response 1:
We sincerely apologize for the insufficient methodological details regarding stool specimen collection in our previous response.
To clarify, the fecal sampling procedure was temporally synchronized with fasting venous blood collection, with both biospecimens acquired during morning hours (08:00-11:00). This temporal synchronization ensured a minimum 10-hour fasting period prior to defecation, thereby minimizing potential confounding effects from breakfast consumption on gut microbial profiles. Dietary patterns were systematically assessed using a validated Food Frequency Questionnaire (FFQ) administered in the same day, enabling quantitative comparison of macronutrient intake variations across cohorts. Furthermore, alcohol consumption patterns were rigorously documented through structured interviews, with comparative analyses demonstrating no intergroup differences in drinking behaviors (Table 1). Pharmaceutical interventions were controlled through our multi-tiered exclusion criteria outlined in Section 2.1:
i) Systemic pathologies (e.g., malignancies, cardiopulmonary/renal/hepatic dysfunction),
ii) Neurological/psychiatric disorders with known microbial associations (e.g., traumatic encephalopathy, epilepsy, Parkinsonism),
iii) Use of cognition-modulating pharmaceuticals or nutraceuticals,
iv) Physical impairments precluding functional assessments.
Notably, our experimental design deliberately prioritized ecological validity over strict standardization, as the primary objective focused on characterizing baseline microbiota variations in free-living populations.
“Study participants were required to provide fasting stool specimens simultaneously with venous blood collection during the designated morning session, under standardized fasting conditions (≥10 hours).”[Page 2, Line 87~89]